# Stochasticity of Deterministic Gradient Descent: Large Learning Rate for Multiscale Objective Function

**Lingkai Kong**
School of Mathematics
University of Science and Technology of China
and Georgia Institute of Technology

**Molei Tao**
School of Mathematics
Georgia Institute of Technology
`mtao@gatech.edu`

## Abstract

This article suggests that deterministic Gradient Descent, which does not use any stochastic gradient approximation, can still exhibit stochastic behaviors. In particular, it shows that if the objective function exhibit multiscale behaviors, then in a large learning rate regime which only resolves the macroscopic but not the microscopic details of the objective, the deterministic GD dynamics can become chaotic and convergent not to a local minimizer but to a statistical distribution. In this sense, deterministic GD resembles stochastic GD even though no stochasticity is injected. A sufficient condition is also established for approximating this long-time statistical limit by a rescaled Gibbs distribution, which for example allows escapes from local minima to be quantified. Both theoretical and numerical demonstrations are provided, and the theoretical part relies on the construction of a stochastic map that uses bounded noise (as opposed to Gaussian noise).

## 1 Introduction

Among first-order optimization methods which are a central ingredient of machine learning, arguably the most used is gradient descent method (GD), or rather one of its variants, stochastic gradient descent method (SGD). Designed for objective functions that sum a large amount of terms, which for instance can originate from big data, SGD introduces a randomization mechanism of gradient subsampling to improve the scalability of GD (e.g., Zhang [2004], Moulines and Bach [2011], Roux et al. [2012]). Consequently, the iteration of SGD, unlike GD, is not deterministic even when it is started at a fixed initial condition. In fact, if one fixes the learning rate (LR) in SGD, the iteration does not converge to a local minimizer like in the case of GD; instead, it converges to a statistical distribution with variance controlled by the LR (e.g., Borkar and Mitter [1999], Mandt et al. [2017], Li et al. [2017]). Diminishing LR was thus proposed to ensure that SGD remains as an optimization algorithm (e.g., Robbins and Monro [1951]). On the other hand, more recent perspectives include that the noise in SGD may actually facilitate escapes from bad local minima and improve generalization (see Sec.1.2 and references therein). In addition, non-diminishing LRs often correspond to faster computations, and therefore are of practical relevance[1]. Meanwhile, GD does not need the LR to be small in order to reduce the stochasticity, although in practices the LR is often chosen small enough to fully resolve the landscape of the objective, corresponding to a stability upper bound of $1/L$ under the common $L$-smooth assumption of the objective function.

We consider deterministic GD[2] with fixed large LR, based on the conventional belief that it optimizes more efficiently than small LR. The goal is to understand if large LR works, and if yes, in what sense.

We will show that in a specific and yet not too restrictive setup, if LR becomes large enough (but not arbitrarily large), GD no longer converges to a local minimum but instead a statistical distribution. This behavior bears significant similarities to SGD, including (under reasonable assumptions):

- starting with an arbitrary initial condition, the empirical distribution of GD iterates (collected along discrete time) converges to a specific statistical distribution, which is not Dirac but almost a rescaled Gibbs distribution, just like SGD;

- starting an ensemble of arbitrary initial conditions and evolving each one according to GD, the ensemble, collected at the same number of iterations, again converges to the same almost Gibbs distribution as the number of iteration increases, also like SGD.

Their difference, albeit obvious, should also be emphasized:

- GD is deterministic, and the same constant initial condition will always lead to the same iterates. No filtration is involved, and unlike SGD the iteration is not a stochastic process.

In this sense, GD with large LR works in a statistical sense. One can obtain stochasticity without any algorithmic randomization! Whether this has implications on generalization is beyond the scope of this article, but large LR does provide a mechanism for escapes from local minima. We'll see that microscopic local minima can always be escaped, and sometimes macroscopic local minima too.

## 1.1 Main Results

How is stochasticity generated out of determinism? Here it is due to chaotic dynamics. To further explain, consider an objective function $f : \mathbb{R}^d \to \mathbb{R}$ that admits a macro-micro decomposition

$$f(x) := f_0(x) + f_{1,\epsilon}(x) \tag{1}$$

where $0 < \epsilon \ll 1$, $f_0, f_{1,\epsilon} \in \mathcal{C}^2(\mathbb{R}^d)$, and the microscopic $f_{1,\epsilon}$ satisfies the following conditions.

**Condition 1.** *There exists a bounded nonconstant random variable (r.v.) $\zeta$, with range in $\mathbb{R}^d$ and $\mathbb{E}\zeta = 0$, such that: $\forall \epsilon > 0$ and $\forall x \in \mathbb{R}^d$, there exists a positive measured set $\Gamma_{x,\epsilon} \subset B(0, \delta(\epsilon))$ with $\lim_{\epsilon \downarrow 0} \delta(\epsilon) = 0$, such that the r.v. uniformly distributed on $\Gamma_{x,\epsilon}$, denoted by $Y_{x,\epsilon}$, satisfies $\nabla f_{1,\epsilon}(x + Y_{x,\epsilon}) \xrightarrow{w} -\zeta$ uniformly with respect to $x$ as $\epsilon \to 0$. Assume without loss of generality that $\mathbb{E}\zeta = 0$ (nonzero mean can be absorbed into $f_0$).*

**Notation:** Throughout this paper '$w$' means weak convergence: a sequence of random variables $\{X_n\}_{n=1}^{\infty}$ has a random variable $X$ as its weak limit, if and only if for any compactly supported test function $g \in \mathcal{C}^{\infty}(\mathbb{R}^d)$, $\mathbb{E}g(X_n) - \mathbb{E}g(X) \to 0$ as $n \to \infty$.

**Condition 2.** *$\epsilon \nabla^2 f_{1,\epsilon}$ is uniformly bounded as $\epsilon \to 0$, and $\exists m \in \mathbb{R}$, s.t. for any bounded rectangle $\Gamma \subset \mathbb{R}^d$ whose area $|\Gamma| > 0$, $\mathbb{E}\left[\ln \|\epsilon \nabla^2 f_{1,\epsilon}(U_\Gamma)\|_2\right] \to m$, where $U_\Gamma$ is a uniform r.v. on $\Gamma$.*

**Example 1** (periodic micro-scale). *For intuition, consider a special case where $f_{1,\epsilon} := \epsilon f_1\left(\frac{x}{\epsilon}\right)$ for a periodic $f_1 \in \mathcal{C}^2(\mathbb{R})$. It is easy to check that both conditions are satisfied.*

**Example 2** (aperiodic micro-scale). *Given a $\mathcal{C}^2$ function $F(x_1, x_2, \cdots, x_N) : \mathbb{R}^d \times \mathbb{R}^d \times \cdots \times \mathbb{R}^d \to \mathbb{R}$, that is periodic in each $x_i \in \mathbb{R}^d$, i.e., there exists constant vector $T \in \mathbb{R}^d$ such that $F(x_1, \cdots, x_i, \cdots, x_N) = F(x_1, \cdots, x_i + T, \cdots, x_N)$ for all $x_1, \cdots, x_N$ and $i = 1, \cdots, N$. Then for any $\omega_1, \cdots, \omega_N \in \mathbb{R}$, $f_{1,\epsilon}(x) := \epsilon F(\frac{\omega_1 x}{\epsilon}, \frac{\omega_2 x}{\epsilon}, \cdots, \frac{\omega_N x}{\epsilon})$ satisfies Cond.1 and 2. If the $\omega$'s are nonresonant, meaning that the only solution to $z_1\omega_1 + z_2\omega_2 + \cdots + z_N\omega_N = 0$ for $z_i \in \mathbb{Z}$ is $z_1 = z_2 = \cdots = z_N = 0$, then $f_{1,\epsilon}$ is not periodic. An example is $f_{1,\epsilon} = \epsilon(g_1(x/\epsilon) + g_2(\sqrt{2}x/\epsilon))$ for any 1-periodic $g_1$ and $g_2$.*

**Remark 1.** Cond.1 and 2 generalize and relax the periodic micro-scale requirement. Still required is, intuitively speaking, that every part of the small scale $f_{1,\epsilon}$ appears similar in a weak sense. In the special case of periodic micro-scale, it is easy to see $f_{1,\epsilon} = \mathcal{O}(\epsilon)$, $\nabla f_{1,\epsilon} = \mathcal{O}(1)$ and $\nabla^2 f_{1,\epsilon} = \mathcal{O}(\epsilon^{-1})$. However, after the relaxation of periodicity requirement, it may only be implied that $\nabla f_{1,\epsilon} = \mathcal{O}(1)$ (Cond.1) and $\nabla f_{1,\epsilon} = \mathcal{O}(\epsilon^{-1})$ (Cond.2). Later on, Cond.1 will help connect deterministic and stochastic maps, and Cond.2 will help estimate the Lyapunov exponent so that the onset of chaos can be quantified.

Fig.1 provides an example of $f$. This class of $f$ models objective landscapes that assume certain macroscopic shapes (described by $f_0$), but when zoomed-in exhibit additional small-in-$x$ and $f$ fluctuations (produced by $f_{1,\epsilon}$). Taking the loss function of a neural network as an example, our intuition is that if the training data is drawn from a distribution, the distribution itself produces the dominant macroscopic part of the landscape (i.e., $f_0$), and noises in the training data could lead to $f_{1,\epsilon}$ which corresponds to small and localized perturbations to the loss (see Appendix C and also e.g., Mei et al. [2018], Jin et al. [2018]).

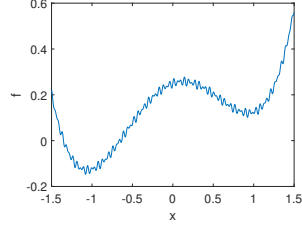

Figure 1: A multi-scale function, $f(x) = (x^2 - 1)^2/4 + x/8 + \epsilon\left(\sin\left(x/\epsilon\right) + \sin\left(\sqrt{2}x/\epsilon\right)\right)$, $\epsilon = 0.01$.

Note although the length and height scales of $f_{1,\epsilon}$ can be both much smaller than those of $f_0$, $\nabla f_0$ and $\nabla f_{1,\epsilon}$ are nevertheless both $\mathcal{O}(1)$, creating nonconvexity and a large number of local minima even if $f_0$ is (strongly) convex.

What happens when gradient decent is applied to $f(x)$, following repeated applications of the map

$$\varphi(x) := x - \eta\nabla f(x) = x - \eta\nabla f_0(x) - \eta\nabla f_{1,\epsilon}(x)?$$

($\eta$ will be called, interchangeably, learning rate (LR) or time step.)

When $\eta \ll \epsilon$, GD converges to a local minimum (or a saddle, or in general a stationary point where $\nabla f = 0$). This is due to the well known convergence of GD when $\eta = o(1/L)$ for $L$-smooth $f$, and $L = \mathcal{O}(\epsilon^{-1})$ for our multiscale $f$'s (Rmk.1).

For $\eta \gg 1$, or more precisely when it exceeds $1/L_0$ for $L_0$-smooth $f_0$, the iteration generally blows up and does not converge. However, there is a regime in-between corresponding to $\epsilon \lesssim \eta \ll 1$, and this is what we call large LR, because here $\eta$ is too large to resolve the micro-scale (i.e., $f_{1,\epsilon}$, whose gradient has an $\mathcal{O}(\epsilon^{-1})$ Lipschitz constant).

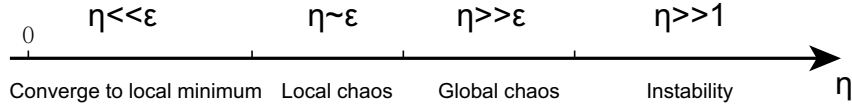

Figure 2: What happens as learning rate increases?

Fig.2 previews what happens over the spectrum of $\eta$ values. The difference between 'local chaos' and 'global chaos' will be detailed in Sec. 2.3.1 and B.3.2.

In fact, for the multiscale function $f$, one may prefer to find a 'macroscopic' local minimum created by $f_0$, instead of being trapped at one of the numerous local minima created by $f_{1,\epsilon}$, which could just be artifacts due to imperfection of training data. A small LR will not be able to do so, but we'll see below that large LR in some sense is better at this: it will lead GD to converge to a distribution peaked at $f_0$'s minimizer(s), despite that the iteration is based on the $\nabla f(x) = \nabla f_0(x) + \nabla f_{1,\epsilon}(x)$.

Our approach for demonstrating the 'stochasticity' of $\varphi$ consists of three key ingredients: (i) construct another map $\hat{\varphi}$, which is a truly stochastic counterpart of $\varphi$, so that they share the same invariant distribution; (ii) find an approximation of the invariant distribution of $\hat{\varphi}$, namely rescaled Gibbs; (iii) establish conditions for $\varphi$ iterations to generate deterministic chaotic dynamics, which provides a route of convergence to a statistical distribution.

More specifically, we define the stochastic map $\hat{\varphi}$ as

$$\hat{\varphi} : x \mapsto x - \eta\nabla f_0(x) + \eta\zeta,$$

where $\zeta$ is defined in Cond.1. Then we have (note many of these results persist in numerical experiments under relaxed conditions; see Sec.3).

**Theorem 1** (informal version of Thm.4). *Fix $\eta$ and let $\epsilon \to 0$. If $\varphi$ has a family of nondegenerate[3] invariant distributions for $\{\epsilon_i\}_{i=1}^{\infty} \to 0$, which converges in the weak sense, then the weak limit is an invariant distribution of $\hat{\varphi}$.*

**Theorem 2** (informal version of Lem.5, Thm.13 & Thm.7). *Suppose $f_0 \in C^2$ is strongly convex and $L$-smooth, and $f_{1,\epsilon} \in C^1$ satisfies condition 1. Then for $\eta \leq C$ with some $C > 0$ independent of $\epsilon$, $\hat{\varphi}$ has an unique invariant distribution, and its iteration converges exponentially fast to this distribution. Moreover, if the covariance matrix of $\zeta$ is isotropic, i.e., $\sigma^2 I_d$, then the rescaled Gibbs distribution $\frac{1}{Z}\exp\left(-\frac{2f_0(x)}{\eta\sigma^2}\right)dx$ is an $\mathcal{O}(\eta^2)$ approximation of it.*

**Theorem 3** (informal version of Thm.8). *Suppose $f_0, f_{1,\epsilon} \in C^1(\mathbb{R})$, $f_0$ is $L$-smooth, grows unboundedly at infinity, and $f_{1,\epsilon}$ satisfies Cond.1. If $f_0$ has a stationary point, then $\exists \eta_J > 0$ such that for any fixed $0 < \eta < \eta_J$, $\exists \epsilon_0 > 0$, s.t. when $\epsilon < \epsilon_0$, the $\varphi$ dynamics is chaotic.*

In addition, we will show the onset of local chaos as $\eta$ increases is via the common route of period doubling [Alligood et al., 1997]. We will also establish and estimate the positive Lyapunov exponent of $\varphi$ in the large LR regime, which is strongly correlated with chaotic dynamics [Lyapunov, 1992].

The reason that we investigate chaos is the following. Although general theories are not unified yet, it is widely accepted that chaotic systems are often ergodic (on ergodic foliations), meaning the temporal average of an observable along any orbit (starting from the same foliation) converges, as the time horizon goes to infinity, to the spatial average of that observable over an invariant distribution (e.g., Eckmann and Ruelle [1985], Young [1998], Ott [2002]). Moreover, many chaotic systems are also mixing (see e.g., Ott [2002]), which implies that if one starts with an ensemble of initial conditions and evolves each one of them by the deterministic map, then the whole ensemble converges to the (ergodic) invariant distribution.

Therefore, our last step in establishing stochasticity of GD is to show the deterministic $\varphi$ map becomes chaotic for large $\eta$. This way, in most situations it is also ergodic and the assumption of Theorem 1 is satisfied, allowing us to demonstrate and quantify the stochastic behavior of deterministic GD. Note that we also know that if $f_0$ has multiple minima and associated potential wells, then GD can have stochastic behaviors with non-unique statistics (see Remark 12, 24 and Section D.5). Therefore, mixing is not provable unless additional conditions are imposed, and this paper only presents numerical evidence (see section 3.1 and D.2). Meanwhile, note (i) since mixing implies ergodicity and Li-Yorke chaos [Akin and Kolyada, 2003, Iwanik, 1991], our necessary conditions are also necessary for mixing, and (ii) proving mixing of deterministic dynamics is difficult, and only several examples have been well understood; see e.g., Sinai [1970], Ornstein and Weiss [1973].

**Remark 2.** For these reasons, we clarify that the theory in this paper does not quantify the speed of convergence of deterministic GD ($\varphi$) to its long time statistical limit. It is only shown that the stochastic map $\hat{\varphi}$ converges to its statistical limit exponentially fast for strongly-convex $f_0$, and the deterministic map $\varphi$ shares the same statistical limit with $\hat{\varphi}$.

**Relevance to machine learning practices:**   see Sec.3.3 (empirical) & C (theoretical) for examples.

## 1.2   Related work

(S)GD is one of the most popular optimizing algorithms for deep learning, not only because of its practical performances, but also due to extensive and profound theoretical observations that it both optimizes well (e.g., Lee et al. [2016], Jin et al. [2017], Du et al. [2019b,a], Allen-Zhu et al. [2019b]) and generalizes well (e.g., Neyshabur et al. [2015], Bartlett et al. [2017], Golowich et al. [2018], Dziugaite and Roy [2017], Arora et al. [2018], Li and Liang [2018], Li et al. [2018], Wei et al. [2019], Allen-Zhu et al. [2019a], Neyshabur and Li [2019], Cao and Gu [2020], E et al. [2020]).

However, to the best of our knowledge, there are not yet many results that systematically study the effects of large learning rates from a general optimization perspective. Jastrzębski et al. [2017] argue that large LR makes GD more likely to avoid sharp minima (we also note whether sharp minima correspond to worse generalization is questionable, e.g., Dinh et al. [2017]). Another result is [Li et al., 2019b], which suggests that large LR resists noises from data. In addition, Smith and Topin [2019] associate large LR with faster training of neural networks. To relate to our work, note it can be argued from one of our results (namely the rescaled Gibbs statistical limit) that LR smooths out shallow and narrow local minima, which are likely created by noisy data. Therefore, it is consistent with [Li et al., 2019b] and complementary to [Jastrzębski et al., 2017] and [Smith and Topin, 2019]. At the same time, one of our contributions is the demonstration that this smoothing effect can be derandomized and completely achieved by deterministic GD. We also note a very interesting recent heuristic observation [Lewkowycz et al., 2020] consistent with our theory (see Fig.2).

Another related result is [Draxler et al., 2018], which suggests that few substantial barriers appear in the loss landscape of neural networks, and this type of landscape fits our model, in which most potential wells are microscopic (i.e., shallow and narrow).

In addition, since we demonstrate stochasticity purely created by large LR, the technique of Polyak-Ruppert averaging [Polyak and Juditsky, 1992] for reducing the variance and accelerating the convergence of SGD is expected to remain effective, even when no stochastic gradient or minibatch approximation is used. A systematic study of this possibility, however, is beyond the scope of this article. Also, our result is consistent with the classical decreasing LR treatment for SGD (e.g., Robbins and Monro [1951]) in two senses: (i) in the large LR regime, reducing LR yields smaller variance (eqn.2); (ii) once the LR drops below the chaos threshold, GD simply converges to a local minimum (no more variance).

Regarding multiscale decomposition (1), note many celebrated multiscale theories assume periodic small scale, (e.g., periodic homogenization [Pavliotis and Stuart, 2008]), periodic averaging [Sanders et al., 2010], and KAM theory [Moser, 1973]). We relaxed this requirement. Moreover, even when Conditions 1,2 fail, our claimed result (stochasticity) persists as numerically observed (see Sec.3.2).

Another important class of relevant work is on continuum limits and modified equations, which Appendix A will discuss in details.

## 2 Theory

*Proofs and additional remarks are provided in Appendix B.*

### 2.1 Connecting the deterministic map and the stochastic map

Here we will connect the stochastic map $\hat{\varphi}$ and the deterministic map $\varphi$. The intuition is that as $\epsilon \to 0$ they share the same long-time behavior. In the following discussion, we fix the learning rate $\eta$, and in order to show the dependence of $\varphi$ on $\epsilon$, we write it as $\varphi_\epsilon$ explicitly in this section.

**Theorem 4** (convergence of the deterministic map to the stochastic map). *Suppose $f_0$ is a L-smooth function and $f_{1,\epsilon}$ satisfies Cond.1. In order to show the dependence of $\varphi$ on $\epsilon$, $\varphi$ is written as $\varphi_\epsilon$ explicitly. Let $\hat{\varphi}(X) := X - \eta \nabla f_0(X) + \eta \zeta$ where $\zeta$ is the r.v. in Cond.1, i.i.d. if $\hat{\varphi}$ is iterated.*

*Assume there exist a set of random variables whose range is in $\mathbb{R}^d$, denoted by $\mathcal{F}$, and a subset $\mathcal{E} \subset \mathbb{R}$ with $0 \in \bar{\mathcal{E}} \backslash \mathcal{E}$, satisfying:*

- *$\varphi_\epsilon$ is continuous in $\mathcal{F}$ in the weak sence $\forall \epsilon \in \mathcal{E}$. Namely, for any r.v. $X \in \mathcal{F}$ and for any sequence of r.v.'s $Y_n : \Omega \to \mathbb{R}^d$ satisfying $\|Y_n\|_\infty := \sup_{\omega \in \Omega} \|Y_n(\omega)\|_2 \to 0$, we have $\varphi_\epsilon(X + Y_n) \xrightarrow{w} \varphi_\epsilon(X)$. (\*)*

*Let $\{\epsilon_i\}_{i=1}^\infty \subset \mathcal{E}$ be a sequence with 0 limit and for each $i$, $X_{\epsilon_i}$ is a fixed point of $\varphi_{\epsilon_i}$. If $X_{\epsilon_i} \xrightarrow{w} X$, then $X$ is a fixed point of $\hat{\varphi}$, i.e., $\hat{\varphi}(X) \overset{w}{=} X$.*

**Remark 3.** In this paper, invariant distributions that are absolutely continuous w.r.t. Lebesgue measure are called to be nondegenerate. Condition (\*) implies nondegeneracy. We ruled out degenerate invariant distributions, which correspond to (convex combinations of) Dirac distributions at stationary points of $f$. In fact, if one starts GD with initial condition that is any stationary point of $f$, GD won't exhibit any true stochasticity no matter how large the LR is. We avoid considering such a degenerate limiting distribution by excluding them from our random variable space.

**Remark 4.** If we further assume that all random variables in $\mathcal{F}$ have uniformly Lipschitz densities, the conclusion can be strengthened due to the sequential compactness of $\bar{\mathcal{F}}$: denote the set of fixed points of $\hat{\varphi}$ by $\hat{\mathcal{P}} \subset \bar{\mathcal{F}}$. Then the set of weak limit points of $\{X_{\epsilon_i}\}_{i=1}^\infty$, denoted by $\mathcal{P} \subset \bar{\mathcal{F}}$, is non-empty, and $\mathcal{P} \subset \hat{\mathcal{P}}$.

### 2.2 The stochastic map: quantitative ergodicity

This section will show that, when $f_0$ is strongly convex, the stochastic map $\hat{\varphi}$ induces a Markov process that is geometric ergodic, meaning it converges exponentially fast to a unique invariant distribution. We will also show that when $\zeta$ is isotropic, the invariant distribution can be approximated

by a rescaled Gibbs distribution. As an additional remark, we also believe that rescaled Gibbs approximates the invariant distribution when $f_0$ is not strongly convex, even though no proof but only numerical evidence is provided (Sec.3.1); however, geometric ergodicity can be lost.

**Lemma 5** (geometric ergodicity). *Consider $\hat{\varphi}(x) = x - \eta\nabla f_0(x) + \eta\zeta$, where $\zeta$ is a bounded random variable in $\mathbb{R}^d$ with 0 mean, i.i.d. if $\hat{\varphi}$ is iterated. If $f_0$ is strongly convex and $L$-smooth, then there exists $\eta_0 \in \mathbb{R}^+$, such that when $\eta < \eta_0$, the map $X \mapsto \hat{\varphi}(X)$ has a unique invariant distribution and the iteration $\hat{\varphi}^{(n)}(X)$ converges (as $n \to \infty$) to the invariant distribution in Prokhorov metric exponentially fast for any initial condition.*

**Proposition 6** (rescaled Gibbs nearly satisfies the invariance equation). *Suppose $f_0 \in \mathcal{C}^1(\mathbb{R}^d)$ is $L$-smooth. Consider $\hat{\varphi}$ defined in Lemma 5. Suppose $\zeta$ is isotropic, i.e. with covariance matrix $\sigma^2 I_d$ for a scalar $\sigma$. Let $X_0$ be a random variable following rescaled Gibbs distribution*

$$X_0 \sim \frac{1}{Z} \exp\left(-\frac{2f_0(x)}{\eta\sigma^2}\right) dx \tag{2}$$

*Then for any $h \in \mathcal{C}^2$ with compact support, we have, for small enough $\eta$, that*

$$\mathbb{E}h(\hat{\varphi}(X_0)) - \mathbb{E}h(X_0) = \mathcal{O}(\eta^3)$$

**Theorem 7** (rescaled Gibbs is an approximation of the invariant distribution). *Assume $f_0 \in \mathcal{C}^2$ is strongly convex and $L$-smooth, and $\zeta$ is isotropic. Consider $\eta < \eta_0$ and denote by $\rho_\infty$ the density of the unique invariant distribution of $\hat{\varphi}$, whose existence and that of $\eta_0$ are given by Lemma 5, then we have, in weak-\* topology,*

$$\rho_\infty = \tilde{\rho} + \mathcal{O}(\eta^2) \tag{3}$$

*where $\tilde{\rho}$ is rescaled Gibbs distribution with density $\tilde{\rho}(x) = \frac{1}{Z}\exp\left(-\frac{2f_0(x)}{\eta\sigma^2}\right)$.*

## 2.3 Deterministic map

Since we want to link the invariant distributions of the deterministic map and the stochastic map, the existence of nondegenerate invariant distribution of the deterministic map (which is important, see Rmk.3) should be understood, as well as the convergence towards it. The last part of Sec.1.1 discussed that chaos can usually provide these properties, but it is not guaranteed, and mathematical tools are still lacking. Thus, in previous theorems, such existence was assumed instead of being proved. We first present two counter-examples to show that nondegenerate invariant distribution can actually be nonexistent. Details will be given in Thm. 16 and 17. Both counter-examples are based on $f_{1,\epsilon} = \epsilon f_1(x/\epsilon)$ for some periodic $f_1$:

1. In 1-dim, for any $f_1 \in \mathcal{C}^2(\mathbb{R})$ and $\epsilon$, $\exists$ a convex $\mathcal{C}^2$ $f_0$ and an $\eta$ arbitrarily large, s.t. any orbit of $\varphi$ is bounded, but the invariant distribution has to be a fixed point (Thm.16)

2. In 1-dim, for any $f_0 \in \mathcal{C}^2(\mathbb{R})$ and $\eta$, $\exists$ a periodic $\mathcal{C}^2$ $f_1$ and an $\epsilon$ arbitrarily small, s.t. any orbit of $\varphi$ is bounded, but the invariant distribution has to be a fixed point (Thm.17)

Then we show GD iteration is chaotic when LR is large enough (for nondegenerate $x_0$).

### 2.3.1 Li-Yorke chaos

In this section, we fix $\eta$ in order to bound the small scale effect in simpler notations, and write the dependence of $\varphi$ on $\epsilon$ explicitly. The main message is $\varphi$ induces chaos in Li-Yorke sense. Note there are several definitions of chaos (e.g. Block and Coppel [2006], Devaney [2018], Li and Yorke [1975], and Aulbach and Kieninger [2001] is a review of their relations). We quote Li-Yorke's celebrated theorem (Li and Yorke [1975]; see also Sharkovskiĭ [Original 1962; Translated 1995]) as Thm. 18 in appendix. Then we apply this tool to the GD map $\varphi$:

**Theorem 8** (sufficient condition for deterministic GD to be chaotic). *Suppose $f_0, f_{1,\epsilon} \in \mathcal{C}^1(\mathbb{R})$, $f_{1,\epsilon}$ satisfies Cond.1, and $f_0$ is $L$-smooth, satisfying $f(x) \to +\infty$ when $|x| \to \infty$, $\lim_{x\to+\infty} f'(x) = +\infty$ and $\lim_{x\to-\infty} f'(x) = -\infty$. If $\exists x$ s.t. $\nabla f_0(x) = 0$, then for any fixed $0 < \eta < 1/L$, $\exists \epsilon_0$, s.t. when $\epsilon < \epsilon_0$, $\varphi_\epsilon$ induces chaotic dynamics in Li-Yorke sense.*

**Remark 5.** Here $\eta$ has an upper bound $\eta_J$, because when $\eta$ is too large, the iteration will be unstable and no interval $J$ closed under $\varphi_\epsilon$ exists (see Def. 1). Rmk. 12 gives an example on how $J$ depends on $\eta$.

**Remark 6.** Li-Yorke theory is restricted to 1D and Thm.8 cannot easily generalize to multi-dim. Lyapunov exponent in Sec.2.3.2 however provides a hint and quantification for chaos in multi-dim.

**Remark 7.** The threshold $\epsilon_0$ may be dependent on the stationary point $x$, and thus $\epsilon_0$ obtained from an arbitrary $x$ may not be the largest threshold under which chaos onsets.

**Remark 8.** The threshold $\epsilon_0$ is only for local chaos to happen. In fact, as the proof will show, only very weak conditions are needed because here chaos onsets due to that GD evolving within a microscopic potential well is a unimodal map. See also Appendix.B.3.2.

However, as $\epsilon$ further decreases beyond the threshold, or equivalently as $\eta$ increases, global chaos onsets shortly after. The idea is, when there is only local chaos but not a global one, the empirical distribution of iterations concentrates at a local minimum inside a microscopic well, but its variance grows as $\eta$ increases. Shortly after, the distribution floods over the barriers of this microscopic well, and then local chaos transits into global chaos. Sec.2.3.2 will allow us to see that both local and global chaos happen when $\eta \sim \epsilon$.

### 2.3.2 Lyapunov exponent

Lyapunov exponent characterizes how near-by trajectories deviate exponentially with the evolution time. A positive exponent shows sensitive dependence on initial condition, is often understood as a lack of predictability in the system (due to a standard argument that initial condition is never measured accurately), and is commonly associated with chaos. Strictly speaking it is only a necessary condition for chaos (see e.g., Strogatz [2018] Chap 10.5), but it quantifies the strength of chaos.

Suppose $(x_0, x_1, ..., x_n, ...)$ is a trajectory of iterated map $\varphi$. Then the following measures the deviation of near-by orbits and thus defines the Lyapunov exponent:

$$\lambda(x_0) = \lim_{n \to \infty} \frac{1}{n} \sum_{i=0}^{n-1} \ln ||\nabla \varphi(x_i)||_2 \qquad (4)$$

This quantity is often independent of the initial condition (see e.g., Oseledec [1968]), and we will see that this is true in numerical experiments with GD. We can quantitatively estimate $\lambda$:

**Theorem 9** (approximate Lyapunov exponent of GD). *Suppose $f_0$ and $f_1$ are both $\mathcal{C}^2$. Suppose the deterministic map is ergodic, and the small scaled effect $f_{1,\epsilon}$ satisfies Cond.2, then the Lyapunov exponent of the deterministic map starting from $x$, denoted by $\lambda(x)$, satisfies*

$$\lim_{\eta \to 0} \lim_{\epsilon \to 0} \left( \lambda(x) - \ln \left( \frac{\eta}{\epsilon} \right) \right) = m,$$

*where $m$ is the constant in Cond. 2.*

*In the special case when $f_1$ is periodic and $f_{1,\epsilon}(x) = \epsilon f_1(x/\epsilon)$, we have, in addition,*

$$\lambda(x) = m + \ln \left( \frac{\eta}{\epsilon} \right) + \mathcal{O}(\epsilon + \eta).$$

**Remark 9.** A necessary condition for chaos is a positive Lyapunov exponent. From $\lambda(x) \approx m + \ln \left( \frac{\eta}{\epsilon} \right)$, we know the threshold for chaos satisfies $\eta > e^{-m} \epsilon$. This threshold does not distinguish between local and global chaos, whose difference was hidden in the higher order term.

## 3 Numerical experiments

*Additional results, such as verifications of statements about chaos (period doubling & Lyapunov exponent estimation), nonconvex $f_0$, gradient descent with momentum, are in Appendix D.*

### 3.1 Stochasticity of deterministic GD: an example with periodic small scale

Here we illustrate that GD dynamics is not only ergodic (on foliation) but also mixing, even when $f_0$ is not strongly convex but only convex (the strongly convex case was proved and will be illustrated in

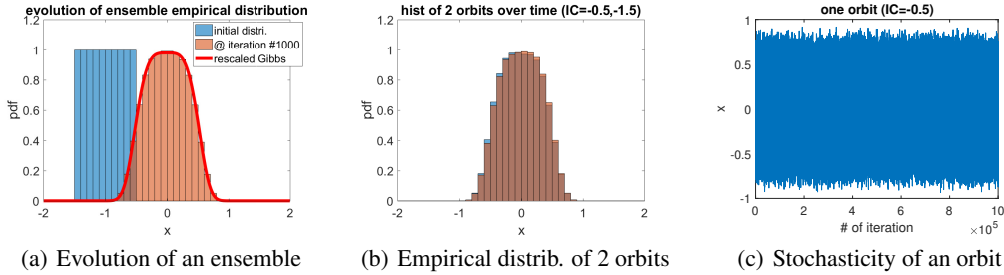

(a) Evolution of an ensemble     (b) Empirical distrib. of 2 orbits     (c) Stochasticity of an orbit

Figure 3: Ergodicity and mixing of $\varphi$. $f_0 = x^4/4$, $f_{1,\epsilon}(x) = \epsilon\sin(x/\epsilon)$ and $\eta = 0.1$, $\epsilon = 10^{-6}$.

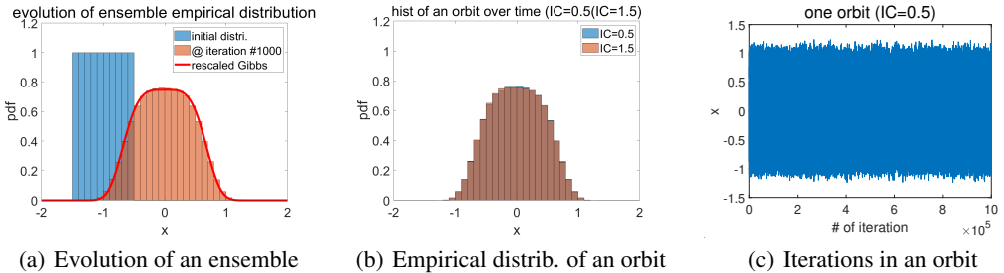

(a) Evolution of an ensemble     (b) Empirical distrib. of an orbit     (c) Iterations in an orbit

Figure 4: Ergodicity and mixing of $\varphi$ for non-periodic $f_{1,\epsilon}$ given in Ex.2 with $\epsilon = 10^{-6}$ and $\eta = 0.1$.

multi-dimension in Appendix D.2). Recall ergodicity is the ability to follow an invariant distribution, and mixing ensures additional convergence to it. Fig.3(a) shows that an arbitrary ensemble of initial conditions converges to approximately the rescaled Gibbs as the number of iteration increases. Fig.3(b) shows the empirical distribution of any orbit (i.e., $x_0, x_1, \cdots$ starting with an arbitrary $x_0$) also converges to the same limit. Fig.3(c) visualizes that any single orbit already appears 'stochastic', even though the same initial condition would lead to exactly the same orbit.

### 3.2 Stochasticity of deterministic GD: two examples with aperiodic small scales

First consider an example whose small scale is not periodic, however satisfying Cond.1 and 2: $f_0 = x^4/4$, $f_{1,\epsilon} = \epsilon\sin(x/\epsilon) + \epsilon\sin(\sqrt{2}x/\epsilon)$. Fig. 4 shows that the system admits rescaled Gibbs as its invariant distribution (Thm. 7) and is ergodic and mixing.

Then we show, numerically, that stochastic behavior of large-LR-GD can persist even when Cond.1 & 2 fail. Here $f_0 = x^2/2$ and $f_{1,\epsilon}(x) = \epsilon\cos(1 + \cos(\frac{\sqrt{3}}{5}x)\frac{x}{\epsilon})$, the former the simplest, and the latter a made-up function that doesn't satisfy Cond.1,2 (due to that $\cos(\frac{\sqrt{3}}{5}x)/\epsilon$ can be 0). See Fig. 5. Note theoretically establishing local chaos (i.e., orbit filling a local potential well of $f_0 + f_{1,\epsilon}$) is still possible, due to unimodal map's universality, e.g., Strogatz [2018]; however, numerically observed is in fact global chaos, in which $f_1$ facilitates the exploration of the entire $f_0$ landscape.

### 3.3 Stochasticity of deterministic GD: a neural network example

To show that stochasticity can still exist in practical problems even when Cond.1,2 are hard to verify, we run a numerical test on a regression problem with a 2-layer neural network. We use a fully connected 5-16-1 MLP to regress UCI Airfoil Self-Noise Data Set [Dua and Graff, 2017], with leaky ReLU activation, MSE as loss, and batch gradient. Fig.6 shows large LR produces stochasticity and Fig.7 shows small LR doesn't, which are consistent with our study.

### 3.4 Persistence of stochasticity when momentum is added to GD

Our theory is only for vanilla gradient decent, but also numerically observed is that deterministic GD with momentum still exhibits stochastic behaviors with large LR. See Appendix D.4.

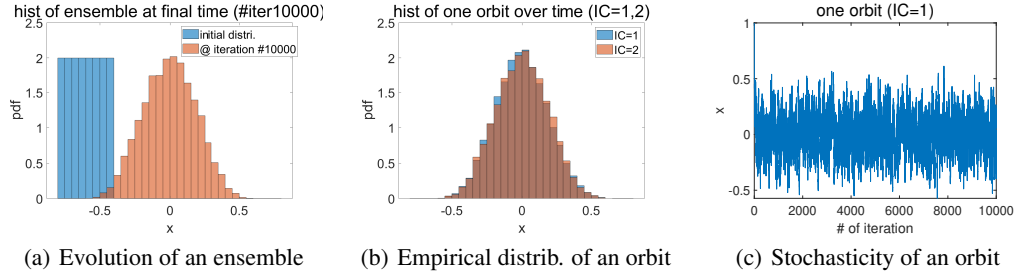

(a) Evolution of an ensemble     (b) Empirical distrib. of an orbit     (c) Stochasticity of an orbit

Figure 5: Ergodicity and mixing of $\varphi$. Nonperiodic nor quasiperiodic small scale. $\epsilon = 10^{-4}, \eta = 0.1$.

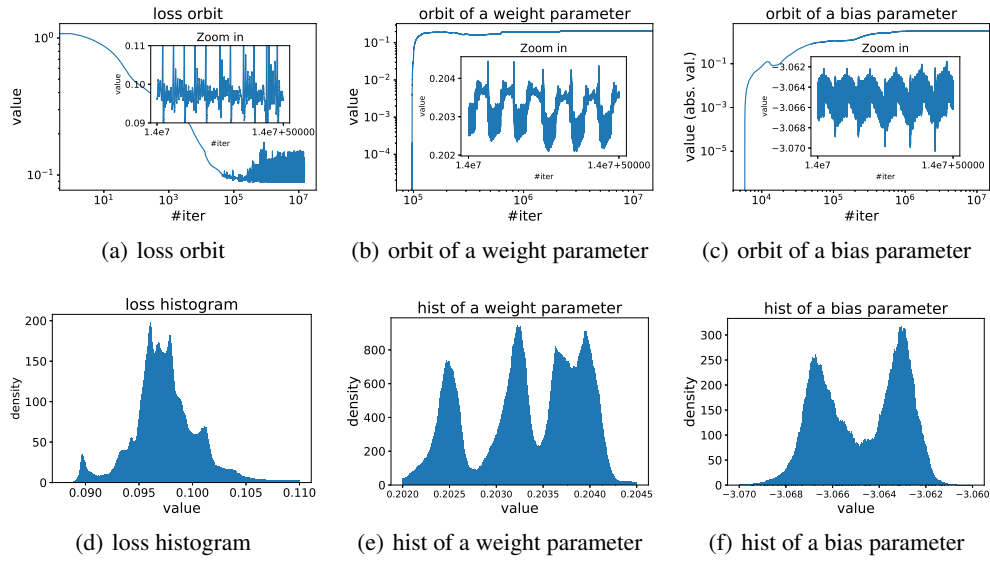

(a) loss orbit     (b) orbit of a weight parameter     (c) orbit of a bias parameter

(d) loss histogram     (e) hist of a weight parameter     (f) hist of a bias parameter

Figure 6: LR=0.02 (large),which demonstrates stochasticity originated from chaos as GD converges to a statistical distribution rather than a local minimum.

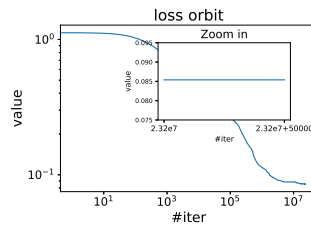

Figure 7: With the same loss function and initial condition, GD with LR=0.0005 (small) converges to a local minimum.

## Broader Impact

This theoretical work deepens our understanding of the performance of gradient descent, an optimization algorithm of significant importance to machine learning. This understanding could lead to the design of better optimization algorithms and improved learning models (either for encouraging or discouraging multiscale landscape, and for enabling or disabling stochasticity originated from determinism, depending on the application). It also helps tune the learning rate, and creates a new quantitative way for generating randomness (more precisely, sampling via determinism). Last but not least, analytical techniques developed and employed in this paper apply to a wide range of other problems.

## Acknowledgments and Disclosure of Funding

This research was mainly conducted when LK was a visiting undergraduate student at Georgia Institute of Technology. The authors thank Jacob Abernethy, Fryderyk Falniowski, Ruilin Li, and Tuo Zhao for helpful discussions. MT was partially supported by NSF DMS-1847802 and ECCS-1936776.

## Footnotes

[1] Optimizing LR is an important subarea but out of our scope; see e.g., Smith [2017] and references therein.

[2] Despite of the importance of SGD, there are still contexts in which deterministic GD is worth studying; e.g., for training with scarce data, for low-rank approximation (e.g., Tu et al. [2015]) and robust PCA (e.g., Yi et al. [2016]), and for theoretical understandings of large neural networks (e.g, Du et al. [2018, 2019b]).

[3]By 'nondegenerate', we require the distribution to be absolutely continuous w.r.t. Lebesgue measure. Invariant distribution of $\varphi$ always exists; an example is a Dirac distribution concentrated at any stationary point of $f$. See Rmk.3.

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
