[Supplementary Material · GDchaos_appendix.pdf]

# A  On the insufficiency of modified equation

Recently there has been an extremely interesting line of research in which discrete algorithms are studied through their continuum limits (e.g., Su et al. [2014], Wibisono et al. [2016], Liu et al. [2017], Franca et al. [2018], Ma et al. [2019], Tao and Ohsawa [2020]); these limits, however, correspond to a small LR (denoted by $\eta$) regime.

It is possible to slightly extend this regime by writing down a limiting ODE that includes additional correction terms (e.g., Shi et al. [2018], Li et al. [2019a], Kovachki and Stuart [2019]). The classical notion for systematically doing so is backward error analysis and modified equation (e.g., Hairer et al. [2006]). For example, the GD map $\varphi$ can be formally approximated, via an application of the modified equation theory, by $\dot{x} = -\nabla \tilde{f}(x)$, where the modified loss

$$\tilde{f}(x) = f(x) + \frac{\eta}{4}\|\nabla f(x)\|_2^2 + \mathcal{O}(\eta^2).$$

While informative, this result does not help us understand the large LR regime. Take $f_{1,\epsilon} = \epsilon f_1(x/\epsilon)$ for periodic $f_1$ as an example. When $\eta \geq C\epsilon$ for some $C > 0$, the formal series expansion used in modified equation does not converge (see Appendix A), which renders it inapplicable.

More precisely, as detailed in Hairer et al. [2006] Chap IX.1, in order for a discrete map

$$\Phi_\eta(x) = x + \eta g(x) \qquad \text{(in our case } g(x) = f'(x) = f_0'(x) + f_1'(x/\epsilon))$$

to be the $\eta$-time flow of

$$\dot{x} = g(x) + \eta g_2(x) + \eta^2 g_3(x) + \cdots, \tag{5}$$

we need

$$g_2(x) = -\frac{1}{2!}g'g(x)$$

$$g_3(x) = -\frac{1}{3!}(g''(g,g)(x) + g'g'g(x)) - \frac{1}{2!}(g'g_2(x) + g_2'g(x))$$

$$\cdots$$

Note each derivative of $g$ gives a factor of $1/\epsilon$, and thus $g_n = \mathcal{O}(\epsilon^{-(n-1)})$. Therefore, RHS of (5) diverges if $\eta \geq C\epsilon$ for some $C > 0$, in which case the more higher-order correction terms are included, the worse approximation power the modified ODE will have.

*This paper thus develops a completely different framework to understand the large LR regime.*

# B  Proofs and additional remarks

## B.1  On the relation between stochastic and deterministic map

**Remark 10** (On Theorem 4).

- The purpose for using an open set $\mathcal{E}$ accumulating at 0 but does not use a interval such as $(0, 1]$ directly here. In the later Theorem 17, we proved that for a fixed $f_0$ and $\eta$, there exists periodic $f_{1,\epsilon}$ and arbitrary small $\epsilon$ to make the non trivial invariant distribution doesn't exist. We can use the set $\mathcal{E}$ to eliminate this bad case that we doesn't want to see.

- Lemma 5 gives a sufficient condition for $\hat{\varphi}$ to have a unique fixed point, denoted by $X$. When this happens, the conclusion will be if $\{X_{\epsilon_i}\}_{i=1}^{\infty}$ has a weak limit, $\{X_{\epsilon_i}\}_{i=1}^{\infty} \to X$. We do numerical tests on this situation in Sec.D.2. When $\hat{\varphi}$ have multiple fixed points, please see related numerical test in Sec.D.5.

- Intuitively, condition (*) means $\varphi_\epsilon$ is continuous in $\mathcal{F}$. This property is used in the proof of lemma 12. Condition (*) is strong, but we can hardly prove it or find a condition that easy to test. The 2-order derative of $f_0$ goes to infinity, which is pathological, but also make the whole problem interesting and nontrivial. See Thm. 16 and 17 for 2 examples. However, some necessary conditions could be useful, such as the r.v.'s in $\mathcal{F}$ cannot have atom points (which means all the variables are nondegenerate).

In order to prove Theorem 4, we need the following lemmas.

**Lemma 10.** *Under the condition of Thm. 4, $\forall X$, there exists $\tilde{X}$, such that $\sup_{\omega \in \Omega} \|\tilde{X}(\omega) - X(\omega)\|_2 < \delta(\epsilon)$ where $\Omega$ is the sample space and $\varphi_\epsilon(\tilde{X}) \xrightarrow{w} \hat{\varphi}(\tilde{X})$ when $\epsilon \to 0$.*

*Proof.* Let $\tilde{X} := X + Y_{X,\epsilon}$, where $Y_{X,\epsilon}$ is defined as in Cond. 1. Without causing confusion, the dependence of $Y_{x,\epsilon}$ on $\epsilon$ is omitted in this proof, as well as in lemma 11 and 12. So $\sup_\omega \|Y_X(\omega)\|_2 < \delta(\epsilon)$. ($\delta(\epsilon)$ is given in Cond. 1)

Arbitrarily choosing a test function $g$, we have

$$\lim_{\epsilon \to 0} \mathbb{E}\left[g(\varphi_\epsilon(\tilde{X})) - g(\hat{\varphi}(\tilde{X}))\right]$$
$$= \lim_{\epsilon \to 0} \mathbb{E}\left[g(\tilde{X} - \eta \nabla f_0(\tilde{X}) - \eta \nabla f_{1,\epsilon}(\tilde{X})) - g(\tilde{X} - \eta \nabla f_0(\tilde{X}) - \eta \zeta)\right]$$
$$= \lim_{\epsilon \to 0} \mathbb{E}_X[\mathbb{E}_{Y_X}[g(X + Y_X - \eta \nabla f_0(X + Y_X)$$
$$- \eta \nabla f_{1,\epsilon}(X + Y_X)) - g(X + Y_X - \eta \nabla f_0(X + Y_X) - \eta \zeta)|X]]$$

We use the nice property of $g$ and $f_0$ to have some of the $Y_X$'s.

$$g(x + Y_x - \eta \nabla f_0(x + Y_x) - \eta \nabla f_{1,\epsilon}(x + Y_x)) = g(x - \eta \nabla f_0(x) - \eta \nabla f_{1,\epsilon}(x + Y_x)) + \mathcal{O}(\delta(\epsilon))$$
$$g(x + Y_x - \eta \nabla f_0(x + Y_x) - \eta \zeta) = g(x - \eta \nabla f_0(x) - \eta \zeta) + \mathcal{O}(\delta(\epsilon))$$

Due to the uniform weak convergence condition in condition 1, we calculate the limit first and then compute the expectation regarding $X$, which means

$$\lim_{\epsilon \to 0} \mathbb{E}\left[g(\varphi_\epsilon(\tilde{X})) - g(\hat{\varphi}(\tilde{X}))\right]$$
$$= \mathbb{E}_X\left[\lim_{\epsilon \to 0} \mathbb{E}_{Y_X}[g(X - \eta \nabla f_0(X) - \eta \nabla f_{1,\epsilon}(X + Y_X)) - g(X - \eta \nabla f_0(X) - \eta \zeta)|X]\right]$$
$$= 0$$

$\square$

**Lemma 11.** *Let $\tilde{X} := X + Y_X$ (as in the proof of Lemma 10). Then $\hat{\varphi}(\tilde{X}) \xrightarrow{w} \hat{\varphi}(X)$ as $\epsilon \to 0$.*

*Proof.* For an arbitrary test function $g$, we have

$$\lim_{\epsilon \to 0} \mathbb{E}\left[g(\hat{\varphi}(\tilde{X})) - g(\hat{\varphi}(X))\right]$$
$$= \lim_{\epsilon \to 0} \mathbb{E}\left[g(\tilde{X} - \eta \nabla f_0(\tilde{X}) - \eta \zeta) - g(X - \eta \nabla f_0(X) - \eta \zeta)\right]$$
$$\leq \lim_{\epsilon \to 0} \mathbb{E}\left[\sup \|\nabla g\| \|(\tilde{X} - \eta \nabla f_0(\tilde{X})) - (X - \eta \nabla f_0(X))\|\right]$$
$$\leq \lim_{\epsilon \to 0} \mathbb{E}\left[\sup \|\nabla g\| (1 + \eta L)\|\tilde{X} - X\|_2\right]$$
$$\leq \lim_{\epsilon \to 0} (1 + \eta L) \sup \|\nabla g\| \, \delta(\epsilon)$$
$$= 0$$

The 3rd last line is due to $L$-smoothness of $f_0$. $\square$

**Lemma 12.** *$\forall X \in \mathcal{F}$, $\varphi_\epsilon(X) \xrightarrow{w} \hat{\varphi}(X)$ when $\epsilon \to 0$.*

*Proof.* We define $\tilde{X} := X + Y_X$, like we did in the proof for lemma 10. Fix a $g$ as the test function.

$$\mathbb{E}\left[g(\varphi_\epsilon(X)) - g(\hat{\varphi}(X))\right]$$
$$= \mathbb{E}\left[g(\varphi_\epsilon(X)) - g(\varphi_\epsilon(\tilde{X}))\right] + \mathbb{E}\left[g(\hat{\varphi}(X)) - g(\hat{\varphi}(\tilde{X}))\right] + \mathbb{E}\left[g(\varphi_\epsilon(\tilde{X})) - g(\hat{\varphi}(\tilde{X}))\right]$$

The first term converges to 0 due to condition (*) in Thm. 4, which ensures the continuity in the weak sense of $\varphi_\epsilon$. The second term goes to 0 according to lemma 11. The third term converges to 0 according to lemma 10. So we have $\mathbb{E}\left[g(\varphi_\epsilon(X)) - g(\hat{\varphi}(X))\right] \to 0$. $\square$

This lemma prepares us to finish the following proof.

*Proof of Thm.4.* Suppose $X_{\epsilon_i} \in \mathcal{F}$ is a sequence of r.v. , which are fixed points for $\varphi_{\epsilon_i}$, and have a limit point $X \in \mathcal{F}$ in the weak sence. Then we have

$$\varphi_\epsilon(X_\epsilon) \overset{w}{=} X_\epsilon, \quad \forall \epsilon = \epsilon_i$$

$$X_{\epsilon_i} \xrightarrow{w} X$$

$$\varphi_{\epsilon_i}(X_{\epsilon_i}) \xrightarrow{w} \hat{\varphi}(X)$$

So $\hat{\varphi}(X) \overset{w}{=} X$. $\qquad\qquad\square$

## B.2 On the stochastic map $\hat{\varphi}$

### B.2.1 Some quantitative results about its ergodicity

*Proof of Lemma 5.* Here we use the machinery provided by Hennion and Hervé [2004]. Regard $\hat{\varphi}$ as a random action on $\mathbb{R}^d$. In this proof, we write the dependence of $\hat{\varphi}$ on $\zeta$ explicitly as $\hat{\varphi}_\zeta$. Choose a fixed point $x_0$ and let

$$c(\zeta) := \sup \left\{ \frac{d(\hat{\varphi}_\zeta x, \hat{\varphi}_\zeta y)}{d(x, y)} : x, y \in \mathbb{R}^d, x \neq y \right\}$$

$$\mathcal{M}_{\gamma+1} := \int_G (1 + c(\zeta) + d(\varphi_\zeta x_0, x_0))^\gamma \, d\pi(\zeta)$$

$$\mathcal{C}_{\gamma+1}^{(n_0)} := \int_G c(\varphi_\zeta) \max\{c(\varphi_\zeta), i\}^\gamma \, d\pi^{*n}(\zeta)$$

In $\hat{\varphi}$ and the our interested chaotic regime of learning rate, since $f_0$ is strongly convex and $L$-smooth, we choose $\eta_0$ small to ensure $c(\varphi_\zeta) = 1 - \eta_0 L < 1$, and we choose $\gamma = 0$, $n_0 = 1$ to get $\mathcal{M}_{\gamma+1} = E_\zeta[1 + c(\varphi_\zeta) + d(\hat{\varphi}_\zeta(x_0), x_0)] < +\infty$ and $\mathcal{C}_{\gamma+1}^{(1)} = E_\zeta[c(\varphi_\zeta)] < 1$.

Under these facts, Theorem 1 in Hennion and Hervé [2004] ensures that there is a unique $\hat{\varphi}$-invariant probability distribution $\hat{\mu}_0$. Moreover, geometric ergodicity holds in the Prokhorov distance $d_P$. Namely, there exists positive real number $C$ and $\kappa_0 < 1$, such that, for any probability distribution $\mu$ on $M$ satisfying $\mu(d(\cdot, x_0)) < +\infty$, and all $n \geq 1$,

$$d_P(\hat{\varphi}_\sharp^{(n)} \mu, \hat{\mu}_0) \leq C \kappa_0^{n/2}$$

where $\hat{\varphi}_\sharp^{(n)}$ stands for apply the push forward of measure $n$ times. $\qquad\qquad\square$

**Remark 11.** In a separable metric space, which is our case, convergence of measures in the Prokhorov metric is equivalent to weak convergence of measures, which is also equivalent to the convergence of cumulative distribution functions.

The following two remarks show that convexity and $L$-smoothness of $f_0$ are necessary for geometric ergodicity established by Lemma 5.

**Remark 12.** Here we will explain in 1-dim, what can happen when the function $f_0$ is not convex. Since the random variable $\zeta$ is bounded, denote it by $[a, b]$. Unlike in a standard overdamped Langevin case, there can be potential barriers in $f_0$ that $\hat{\varphi}$ cannot cross, because the noise is of a finite strength. To make this quantitative, we assume the existence of an invariant distribution with density $\mu_0$, and calculate what kind of points are not in the support of $\mu_0$. When $\eta < 1/L$, for a point $x \in \text{supp}\hat{\mu}_0$, we have $\eta f_0'(x) \in \eta[a, b]$. So if $\{x | f_0'(x) \in [a, b]\}$ is not a connected set (note that it is independent from $\eta$), then the support of the invariant density will be separated in to disjoint components, and no orbit can jump between them. An example explains why the set can be disconnected:

Suppose $f_0 = k(x^2 - 1)^2$, $k > 0$ for example, and $f_{1,\epsilon} = \epsilon \sin(x/\epsilon)$. Calculate the set $S := \{x : f_0'(x) \in [-1, 1]\} = \{x : |4kx(x^2 - 1)| < 1\}$. We have that when $k < \frac{3\sqrt{3}}{8}$, $S$ is connected. But when $k > \frac{3\sqrt{3}}{8}$, the set $S$ is not connected. In this case, a point cannot jump from one well to another as $\hat{\varphi}$ is closed in each connected component of $S$, which means ergodicity on $S$ is lost. Which distribution the system converges to (if existent) relies on which well the initial condition belongs to.

In multi-dimension case, connectedness is different from simply connectedness, which complicates the intuition. We won't discuss it here.

See also Sec. D.5 on jumping between potential wells by the deterministic map.

**Remark 13.** When $f_0$ is not $L$-smooth, such as $f_0(x) = (x^2 + 1)^2$ and $f_{1,\epsilon} = \epsilon \sin(x/\epsilon)$. For a fixed $\eta$, it is easy to see that when the absolute value of initial condition is greater than $x_0$, where $x_0$ is the greatest solution of $x - 4\eta x(x^2 + 1) + \eta + x = 0$, we know $P(|\hat{\varphi}(x)| > |x|) = 1$, so the system will explode and never converge to any distribution. This is because $\mathcal{M}_{\gamma+1} < \infty$ in the proof of Lemma 5 is not satisfied.

**Theorem 13** (coupling estimation of the exponential convergence rate of $\hat{\varphi}$). *Consider the iteration $x_{k+1} = x_k - \eta \nabla f_0(x_k) + \eta \zeta_k$ for i.i.d. $\zeta_k \sim \zeta$. Denote by $\rho_k$ the density of $x_k$. Assume $f_0$ is $\mathcal{C}^2$, $\nu$-smooth and $\mu$-strongly convex, and $f_1$ is $\mathcal{C}^1$. Then the limiting distribution $\rho_\infty$ exists and the 2-Wasserstein distance satisfies the nonasymptotic bound*

$$W_2(\rho_k, \rho_\infty) \leq (\max\{|1 - \eta\mu|, |1 - \eta\nu|\})^k C \tag{6}$$

*for some constant $C \geq 0$.*

*Proof.* Existence of $\rho_\infty$ is guaranteed by Lemma 5.

Let $\hat{x}_0$ be a random variable distributed according to $\rho_\infty$ and define

$$\hat{x}_{k+1} = \hat{x}_k - \eta \nabla f_0(\hat{x}_k) + \eta \zeta_k$$

using the same noise $\zeta_k$. Then

$$x_{k+1} - \hat{x}_{k+1} = x_k - \hat{x}_k - \eta \left( \nabla f_0(x_k) - \nabla f_0(\hat{x}_k) \right)$$

Since $f_0$ is $\mathcal{C}^2$, $\nu$-smooth and $\mu$-strongly convex, it is easy to see that the mapping $x \mapsto x - \eta \nabla f_0(x)$ is a contraction with rate$= \max\{|1 - \eta\mu|, |1 - \eta\nu|\}$. Therefore,

$$\|x_{k+1} - \hat{x}_{k+1}\| \leq \max\{|1 - \eta\mu|, |1 - \eta\nu|\}\|x_k - \hat{x}_k\|$$

Thus,

$$\mathbb{E}\|x_{k+1} - \hat{x}_{k+1}\|^2 \leq \max\{|1 - \eta\mu|, |1 - \eta\nu|\}^{2k} \mathbb{E}\|x_0 - \hat{x_0}\|^2$$

Note $\hat{x}_k$ is distributed according to $\rho_\infty$ because that is the invariant distribution and $\hat{x}_0 \sim \rho_\infty$. By definition,

$$W_2(\rho_k, \rho_\infty)^2 = \inf_{\pi \in \Pi(\rho_k, \rho_\infty)} \int \|y_1 - y_2\|^2 d\pi(y_1, y_2)$$
$$\leq \mathbb{E}\|x_k - \hat{x}_k\|^2.$$

Therefore, the choice of $C = \sqrt{\mathbb{E}\|x_0 - \hat{x}_0\|^2}$ leads to eq.6. $\qquad\square$

**Corollary 14** (Spectral gap of $\hat{\varphi}$ is at least at the order of $\eta$). *Consider the setup of Thm.13 and $\eta < \frac{1}{\nu}$. Denote by $L$ the transition operator of the Markov process generated by $\hat{\varphi}$, i.e., $L\rho_k = \rho_{k+1} \quad \forall k$. Then $L$ has a single eigenvalue of 1, and any other eigenvalue $\lambda$ satisfies $|1 - \lambda| \geq \eta\mu$.*

*Proof.* Since $\hat{\varphi}$ generates a Markov process, any eigenvalue has modulus bounded by 1.

The single eigenvalue of 1 is guaranteed by geometric ergodicity (Lemma 5). Thus, for any other eigenvalue $\lambda$, $|\lambda| < 1$.

Let $\rho_\perp$ be the eigenfunction corresponding to $\lambda$. Since $L$ preserves the normalization of probability density, $\int \rho_\perp = 0$.

For any $\alpha \neq 0$, let $x_0$ be a random variable distributed according to density $\rho_\infty + \alpha\rho_\perp$. We have

$$\rho_{x_k} = L^k(\rho_\infty + \alpha\rho_\perp) = \rho_\infty + \alpha\lambda^k\rho_\perp$$

and therefore the $L_1$ distance satisfies

$$d_1(\rho_{x_k}, \rho_\infty) = \alpha\lambda^k\|\rho_\perp\|_1$$

Since densities exist, we have the total variation distance

$$d_{TV}(\rho_{x_k}, \rho_\infty) = \frac{1}{2}d_1(\rho_{x_k}, \rho_\infty) = \frac{1}{2}\alpha\|\rho_\perp\|_1\lambda^k$$

Although in general total variation distance cannot be upper bounded by Wasserstein distance, it was shown in Chae et al. [2017] Lemma 5.1 that such an upper bound exists when both probability distributions admit smooth densities, i.e.,

$$d_{TV}(\rho_{x_k}, \rho_\infty) \leq C W_2(\rho_{x_k}, \rho_\infty)$$

for some $C \geq 0$. Combined with Thm. 13, this thus gives

$$d_{TV}(\rho_{x_k}, \rho_\infty) \leq \hat{C} \left( \max\{|1 - \eta\mu|, |1 - \eta\nu|\} \right)^k$$

for some $\hat{C} \geq 0$. Therefore, $|\lambda| \leq \max\{|1 - \eta\mu|, |1 - \eta\nu|\} = 1 - \eta\mu$ (the last equality is due to $\mu \leq \nu$ and $\eta < 1/\nu$). This leads to $|1 - \lambda| \geq \eta\mu$. $\qquad\square$

### B.2.2 On Proposition 6

To prove the bound of difference between $\mathbb{E}h(\hat{\varphi}(X_0))$ and $\mathbb{E}h(X_0)$, we first prove the following lemma:

**Lemma 15** (gradient estimate of rescaled Gibbs). *Suppose $f_0$ is L-smooth. Let $x_0$ be the global minimizer of $f_0$. If*

$$f_0(x) - f_0(x_0) \geq C_1 \|x - x_0\|^{k_1} \text{ and } \|\nabla f_0(x)\| \leq C_2 \|x - x_0\|^{k_2}, \quad \forall x \in \mathbb{R}^d,$$

*Then we have, for $X_0$ following rescaled Gibbs (2),*

$$\mathbb{E}\|\nabla f_0(X_0)\|_2^2 = \mathcal{O}(\eta^{\frac{2k_2-1}{k_1}}) \quad \text{when } \eta \to 0.$$

.

*Proof.*

$$
\begin{aligned}
\mathbb{E}\|\nabla f_0(X_0)\|_2^2 &= \frac{1}{Z_1} \int \|\nabla f_0(x)\|_2^2 \exp\left(-\frac{2f_0(x)}{\eta}\right) dx \\
&\leq \frac{\sqrt[k]{\eta}}{Z_2} \int \|\nabla f_0(x)\|_2^2 \exp\left(-2C_1 \left(\frac{\|x\|}{\sqrt[k_1]{\eta}}\right)^{k_1}\right) d\frac{x}{\sqrt[k]{\eta}} \\
&= \frac{\sqrt[k_1]{\eta}}{Z_2} \int \|\nabla f_0(\sqrt[k_1]{\eta}u)\|_2^2 \exp(-2C_1 \|u\|^{k_1}) du
\end{aligned}
$$

Since

$$\|\nabla f_0(x)\| \leq C_2 \|x - x_0\|^{k_2}$$

So

$$
\begin{aligned}
\mathbb{E}\|\nabla f_0(Y_0)\|_2^2 &= \frac{\sqrt[k_1]{\eta}}{Z_4} \int C_2(\sqrt[k_1]{\eta}\|u\|)^{2k_2} \exp(-2C_1\|u\|^{k_1}) du \\
&= \eta^{\frac{2k_2-1}{k_1}} \frac{1}{Z_4} \int C_2\|u\|^{2k_2} \exp(-2C_1\|u\|^{k_1}) du
\end{aligned}
$$

The integral converges and is a constant, so we have

$$\mathbb{E}\|\nabla f_0(X_0)\|_2^2 = \mathcal{O}(\eta^{\frac{2k_2-1}{k_1}})$$

$\qquad\square$

*Proof of Prop. 6.* Because $\tilde{\zeta}$ is compactly supported and $\|\nabla f_0\|$ is bounded, Taylor expansion of $h$ in $\eta$ gives, $\forall X$,

$$
\begin{aligned}
\mathbb{E}(h(\hat{\varphi}(X))) &= \mathbb{E}_X \left[ \mathbb{E}_{\tilde{\zeta}}[h(X - \eta\nabla f_0(X) + \eta\tilde{\zeta})|X] \right] \\
&= \mathbb{E}_X h(X - \eta\nabla f_0(X)) + \eta\mathbb{E}\tilde{\zeta}^\top \mathbb{E}_X \left[ \nabla h(X - \eta\nabla f_0(X)) \right] \\
&\quad + \frac{\eta^2}{2} \mathbb{E}_X \left[ \mathbb{E}_{\tilde{\zeta}}[\tilde{\zeta}^\top \text{Hess } h(X - \eta\nabla f_0(X))\tilde{\zeta}|X] \right] + \mathcal{O}(\eta^3) \\
&= \mathbb{E}_X \left[ h(X) - \eta\nabla f_0(X)^\top \cdot \nabla h(X) + \frac{\eta^2}{2} \nabla f_0(X)^\top \text{Hess } h(X)\nabla f_0(X) + \frac{\eta^2}{2}\mathbb{E}\tilde{\zeta}^\top \text{Hess } h(X)\mathbb{E}\tilde{\zeta} \right] + \mathcal{O}(\eta^3)
\end{aligned}
$$

When $X = X_0$, we first estimate the 3rd term. Since Hess$h$ is bounded and due to the $L$-smoothness and strong convexity of $f_0$, we know it is $\mathcal{O}(\eta^3)$ using Lemma 15 in the case $k_1 = k_2 = 2$. So we get

$$\mathbb{E}(h(\hat{\varphi}(X_0))) - \mathbb{E}h(X_0)$$
$$= \frac{\eta^2}{2Z} \int \left[ -\frac{2}{\eta} \nabla f_0(x)^\top \cdot \nabla h(x) + \sigma^2 \mathrm{Tr}\, \mathrm{Hess}\, h(x) \right] \exp\left( -\frac{2 f_0(x)}{\eta \sigma^2} \right) dx + \mathcal{O}(\eta^3)$$

And then we use Stokes' theorem to prove the integration in RHS vanishes. Denote

$$\omega := \sum_i (-1)^i \nabla_i h(x) \exp\left( -\frac{2 f_0(x)}{\eta \sigma^2} \right) dx_1 \wedge \cdots \wedge \widehat{dx_i} \wedge \cdots \wedge dx_n$$

where $\widehat{dx_i}$ means dropout $dx_i$. Then

$$d\omega = \sum_i \nabla_i^2 h(x) \exp\left( -\frac{2 f_0(x)}{\eta \sigma^2} \right) - \frac{2}{\eta \sigma^2} \nabla_i h(x) \nabla_i f_0(x) \exp\left( -\frac{2 f_0(x)}{\eta \sigma^2} \right) dx_1 \wedge ... \wedge dx_n$$

$$= \left( \mathrm{Tr}\, \mathrm{Hess}\, h - \frac{2}{\eta \sigma^2} \nabla h^\top \cdot \nabla f_0 \right) \exp\left( -\frac{2 f_0(x)}{\eta \sigma^2} \right) dx_1 \wedge \cdots \wedge dx_n$$

According Stokes' formula,

$$\mathbb{E}(h(\hat{\varphi}(X))) - \mathbb{E}h(X) = \frac{\eta^2 \sigma^2}{2Z} \int_{\mathbb{R}^d} d\omega + \mathcal{O}(\eta^3)$$
$$= \frac{\eta^2 \sigma^2}{2Z} \lim_{r \to \infty} \int_{B(0,r)} d\omega + \mathcal{O}(\eta^3)$$
$$= \frac{\eta^2 \sigma^2}{2Z} \lim_{r \to \infty} \int_{\partial B(0,r)} \omega + \mathcal{O}(\eta^3)$$

The first term vanishes since $h(x)$ is compacted supported, which gives us the conclusion that

$$\mathbb{E}(h(\hat{\varphi}(X_0))) - \mathbb{E}h(X_0) = \mathcal{O}(\eta^3)$$

$\square$

**Remark 14.** Note that strong convexity and $L$-smoothness of $f_0$ are sufficient to satisfy the condition of Lemma 15, but they may not be necessary. In fact, Prop. 6 is also correct for any $f_0$ that satisfies

$$f_0(x) - f_0(x_0) \geq C_1 \|x - x_0\|^{k_1} \text{ and } \|\nabla f_0(x)\| \leq C_2 \|x - x_0\|^{k_2}, \quad \forall x \in \mathbb{R}^d,$$

where $2k_2 - 1 \geq k_1$. Although we only proved that the rescaled Gibbs approximates the invariant distribution when $f_0$ is strongly convex functions, the fact that rescaled Gibbs nearly satisfies the invariance equation does not require strong convexity. In fact, we conjecture that rescaled Gibbs also approximates the invariant distribution for convex and even nonconvex $f_0$. See numerics in Sec.3.1 ($f_0 = x^4/4$, with $k_1 = 4$, $k_2 = 3$) and Appendix D.5 (nonconvex and multimodal $f_0$).

### B.2.3   On Theorem 7

*Proof.* Denote (as before) by $L$ the transition operator of the Markov process generated by $\hat{\varphi}$. Consider a deviation function

$$r := \rho_\infty - \tilde{\rho}.$$

Decompose $r$ as an orthogonal sum

$$r = r_1 + r_0 \quad \text{where } r_1 \in \ker(I - L) \text{ and } r_0 \perp \ker(I - L)$$

Since $\hat{\varphi}$ induces a geometric ergodic process, $\dim \ker(I - L) = 1$, and thus

$$r = \gamma \rho_\infty + r_0 \quad \text{for some scalar } \gamma.$$

Since $L\rho_\infty = \rho_\infty$ and $L\tilde{\rho} = \tilde{\rho} + \mathcal{O}(\eta^3)$ (Prop.6; note weak-* topology is metrizable on a separable space), we have $(I - L)r = \mathcal{O}(\eta^3)$, and consequently

$$(I - L)r_0 = \mathcal{O}(\eta^3)$$

Since $r_0$ is orthogonal to $\ker(I - L)$ which is the eigenspace associated with eigenvalue 1 of $L$, and all eigenvalues of $I - L$, except for the the irrelevant 0, satisfy $|\lambda| \geq \mu\eta$ due to Cor.14, we obtain

$$r_0 = \mathcal{O}(\eta^2).$$

This means $\rho_\infty - \tilde{\rho} = \gamma\rho_\infty + \mathcal{O}(\eta^2)$. Since $\rho_\infty$ and $\tilde{\rho}$ are both density functions that normalize to 1, applying a uniform test function and letting its support go to infinity give $0 = \gamma + \mathcal{O}(\eta^2)$. This yields eq.3. $\qquad\square$

**Remark 15.** The invariant distribution can be approximated by not only rescaled Gibbs but a Gaussian if $f_0$ is strongly convex. Here is the intuition of a more general result:
Consider rescaled Gibbs (2). Due to the small $\eta$ at the denominator, $X_0$ assumes small values with exponentially large probability. We thus can formally Taylor expand $f_0(x)$ about $x = 0$, which we assumed WLOG to be the minimizer. Denote the first nonzero derivative of $f_0$ at 0 by the $k$th one. Then $f_0(x) \approx \frac{1}{k!}f_0^k(0)x^k$. So, from the density of rescaled Gibbs, we see the density of $\frac{X_0}{\sqrt[k]{\eta}}$ can be approximated by

$$\frac{X_0}{\sqrt[k]{\eta}} \sim \frac{1}{Z}\exp\left(\frac{-2f_0^k(0)}{k!\sigma^2}x^k\right)$$

Note that iff $f_0$ is strongly convex, $k = 2$, and one gets a Gaussian approximation.

**Remark 16.** If one considers another stochastic map $\tilde{\varphi}(x) := x - \eta\nabla f_0(x) + \eta\sigma\xi$ where $\xi$ is standard i.i.d. Gaussian, $\tilde{\varphi}(x)$ admits, under the same Lipschitz and convexity conditions, a similar limiting invariant distribution $\frac{1}{Z}\exp\left(-\frac{2f_0(x)}{\eta\sigma^2}\right)$ will be obtained. The key difference is, unlike $\tilde{\varphi}$ which uses unbounded noise and is the discretization of an SDE, our stochastic map $\hat{\varphi}$ uses only bounded noise as it mimics the deterministic map $\varphi$.

### B.3 On the deterministic map $\varphi$

#### B.3.1 counter-examples

Here are the complete version of the 2 counter-examples given in Sec. 2.3.

**Theorem 16** (a sufficient condition for the nonexistence of nondegenerate invariant distribution). *When $d = 1$, for any fixed $\epsilon$ and fixed periodic $f_1 \in \mathcal{C}^2(\mathbb{R})$, for any $\eta_0$, there exists $\eta > \eta_0$ and $f_0 \in \mathcal{C}^2$ such that $|f_0'|$ and $|f_0''|$ (but 3-order or more derivative will explode) are arbitrarily small. For such $f_0$, the orbit starting at any point is bounded but $\varphi$ does not admit a nontrivial invariant distribution.*

*Proof.*

$$\varphi'(x) = 1 - \eta f_0''(x) - \frac{\eta}{\epsilon}f_1''\left(\frac{x}{\epsilon}\right)$$

Because of the continuity of $f_1''$, $1 - \frac{\eta}{\epsilon}f_1''(\frac{x}{\epsilon})$ has a zero point, denote as $x_0$. So we can choose $\delta$ to make $\frac{1-\eta/\epsilon f_1''(x/\epsilon)}{\eta}$ arbitrarily small on the interval $I = [x_0 - \delta, x_0 + \delta]$. Then construct $f_0|_I$ and $\eta$ making $\varphi' \equiv 0$ on $I$. After that, we adjust $f_0$ to make $\varphi(x_0)$, which is not in $I$, be a fixed point of $\varphi$. According to the property of Li-Yorke chaos, all the point will be finally mapped to $I$, and then to $\varphi(x_0)$ and never move. So the nontrivial invariant distribution does not exist. $\qquad\square$

**Theorem 17** (another sufficient condition for the nonexistence of invariant distribution). *When $d = 1$, $\forall$ fixed $f_0 \in \mathcal{C}^2$ and $\eta > 0$, there exists periodic $f_1 \in \mathcal{C}^2$ whose period is 1 and 0,1,2-order derivative is arbitrary small, together with an $\epsilon$ arbitrarily small, making nontrivial invariant distribution not exist.*

*Proof.* Choose $f_1$ s.t. $\nabla^2 f_1(\frac{x}{\epsilon}) \equiv \frac{\epsilon}{\eta}(1 - \eta\nabla^2 f_0(x))$ on a interval $[0, \delta]$ where $\delta \ll \epsilon$ and make $f_1$ and $f_1'$ arbitrarily small on $[0, \delta/\epsilon]$, and choose $f_1$ on $[\delta/\epsilon]$ to ensure continuity and smoothness. We can make $\epsilon \to 0$ to make $f_1''$ small. Then choose a specific $\epsilon$ to make $\varphi(0)$ is a fix point. According to the property of Li-Yorke chaos, all the point will be finally mapped to $[0, \delta]$, then to $\varphi(0)$ and never move. So the nontrivial invariant distribution does not exist. $\qquad\square$

**Remark 17.** The requirements for $\eta$ to be arbitrarily large in Theorem 16 and $\epsilon$ to be arbitrarily small in Theorem 17 ensure the system won't converge to a local minimum created by $f_1$, and from the construction of the counter-examples, we know the system is not the other trivial one, which means the system explodes because $\eta$ is too large.

**Remark 18.** Here we give some intuition of Thm.16 and 17. Thm.18 will show that in 1-dim case, if we have a period-3 orbit, then there exists a subset $S$ of the whole space $J$ satisfying: For every $x_1, x_2 \in S$ with $x_1 \neq x_2$, $\liminf_{n\to\infty} |\varphi^{(n)}(x_1) - \varphi^{(n)}(x_2)| = 0$. So the intuition for proving Thm. 16 and 17 is to make $\varphi \equiv 0$ on a small interval, then all the points that drop in this interval will be mapped to a single fixed point of $\varphi$.

### B.3.2 Period Doubling

When $\eta$ is small, each (local) minimizer of $f$ corresponds to a stable fixed point of $\varphi$, which is thus also a periodic orbit of $\varphi$ with period 1. As $\eta$ increases, this point remains as a fixed point but will become unstable. Instead, the previously stable periodic orbit bifurcates into a stable periodic orbit with period 2, and the period similarly keeps doubling as $\eta$ further increases. Eventually, the period becomes arbitrarily large before a finite value of $\eta$, as will be numerically illustrated in Sec.9. This phenomenon is known as period doubling, which is a common route to chaos (e.g., Alligood et al. [1997], Ott [2002]); after the appearance of arbitrarily large period, the system enters $\eta$ regime that corresponds to chaotic dynamics.

We now explain how this relates to what we call global and local chaos, which are specific to our multiscale problem.

When $\eta \ll \epsilon$, we know GD converges to a local minimum of $f$ corresponding to one of the many potential wells of created by $f_{1,\epsilon}$. This is the non-interesting case.

When $\eta$ approaches some order function of $\epsilon$ describing the width of microscopic potential wells of $f_{1,\epsilon}$ (for the periodic case, this is $\mathcal{O}(\epsilon)$), the orbit is still trapped in a single microscopic potential well, but it starts making jumps within the well. In fact, restricted to any potential well, $\varphi$ becomes a unimodal map (see e.g., Strogatz [2018]) and its dynamics is known to eventually become chaotic as $\eta$ exceeds a critical value. This is where the period of a periodic orbit keeps on doubling and becomes arbitrarily large. The classical method for studying the invariant distribution of unimodal chaotic maps applies here (see e.g., Cvitanovic [2017]). This is the local chaos regime.

Even more interesting is the case when $\eta$ gets even larger, large enough for the orbit to jump out of a single potential well created by $f_{1,\epsilon}$ and navigate the landscape of $f_0$. This is what we call global chaos. For this, Thm.4 and 5 characterize the combined effect of chaos and global behavior of $f_0$.

### B.3.3 About Li-Yorke Chaos

**Definition 1** (Li-Yorke chaos). Let $J$ be an interval and let $F : J \to J$ be continuous. The dynamical system generated by $F$ exhibits Li-Yorke chaos if

1. For any $k = 1, 2, ...$, there is a periodic point in $J$ having period $k$.

2. There is an uncountable set $S \subset J$ containing no periodic points, that satisfies:
   (A) For every $p, q \in S$ with $p \neq q$, $\limsup_{n\to\infty} |F^n(p) - F^n(q)| > 0$ and $\liminf_{n\to\infty} |F^n(p) - F^n(q)| = 0$.
   (B) For every $p \in S$ and periodic point $q \in J$, $\limsup_{n\to\infty} |F^n(p) - F^n(q)| > 0$.

**Theorem 18** (period 3 implies chaos). *If there exists $a \in J$ for which $b = F(a)$, $c = F^2(a)$, and $d = F^3(a)$ satisfy $d \leq a < b < c$ or $d \geq a > b > c$, then $F$ induces Li-Yorke chaos.*

**Remark 19.** About Thm.18, see Sharkovskiĭ [Original 1962; Translated 1995], Li and Yorke [1975] for rigorous theorems and proofs. This is one of the most celebrated result in chaotic dynamics, which tells us that period 3 implies chaos. The 1st conclusion is named after Sharkovskii. The 2nd conlusion in this theorem is also generalized to be the definition of Li-Yorke Chaos in multi-dim case.

Figure 8: Guideline to finding a period-3 orbit

*Proof of Thm.8.* First we show there exists an interval $J$, such that when $0 < \eta < 1/L$, $\varphi(J) \subset J$. WLOG, suppose $f_0(0) = 0$. According to Cond. 1, there exists $\epsilon_1$, when $\epsilon < \epsilon_1$, $\sup_x \|\nabla f_{1,\epsilon}(x)\|$ is uniformly bounded w.r.t. $\epsilon$. Denote the upper bound as $R$. Due to the $L$-smoothness of $f_0$,

$$\limsup_{x \to +\infty}[\varphi(x) - x] \leq \limsup_{x \to +\infty}[-\eta f_0'(x) + \eta R] < -C < 0$$

$$\liminf_{x \to +\infty}[\varphi(x) + x] \geq \liminf_{x \to +\infty}[2x - \eta f_0'(x) + \eta R] \geq \liminf_{x \to +\infty}[(2 - \eta L)x + \eta R] > C > 0$$

where $C > 0$ is a constant. So there exists $M_1$ such that $-x < \varphi(x) < x$ when $x > M_1$. Similarly, we have $M_2$ such that $x < \varphi(x) < -x$ when $x < -M_2$.

So there exists $M := \max(M_1, M_2)$, so when $|x| > M$, $-|x| < \varphi(x) < |x|$. Set $J := [\inf_{x \in [-M,M]} \varphi(x), \sup_{x \in [-M,M]} \varphi(x)]$ and we have $\varphi(J) \subset J$ when $\epsilon < \epsilon_1$.

Next, we try to find $a$, $b$, $c$ and $d$ in Thm. 18. Because $P(\zeta = 0) < 1$, $\exists \delta_0 > 0$ s.t.$P(\zeta > \delta_0) > 0$ and $P(\zeta < -\delta_0) > 0$. Since $\nabla f_0$ have a zero point, we can find an interval $\tilde{J}$ on which $|\nabla f_0| < \delta_0/3$. Denote the middle point of $x_0$. Find a subinterval of $\tilde{J}$, whose length $\leq \eta/\frac{\delta_0}{3}$ and denote as $J$. Divide $J$ into 2 parts of similar length $J_1$ and $J_2$. $\exists \epsilon_1$, s.t. when $\epsilon < \epsilon_1$, $|\min_{J_i} \nabla f_{1,\epsilon}|, |\max_{J_i} \nabla f_{1,\epsilon}| > \frac{2}{3}\delta_0, i = 1, 2$. So now we have that $|\inf_{J_i} \nabla f|, |\sup_{J_i} \nabla f| > \delta_0/3$. Which means we can find $x_1, x_2 \in J_1, x_3, x_4 \in J_2$ and $x_1 < x_2 < x_3 < x_4$ satisfying $\varphi(x_1) = x_1, \varphi(x_2) > x_4, \varphi(x_3) = x_3$, $\varphi(x_4) < x_1$.

Let $c = x_4$, and $d = \varphi(c)$. So we have $\varphi(x_2) > c$. And since $\varphi(x_1) = x_1$ and continuity, $b \in [x_1, x_2]$ s.t.$\varphi(b) = c$. By the same way we get $a \in [x_1, b]$ s.t. $\varphi(a) = b$. Let $\epsilon_0 := \min(\epsilon_1, \epsilon_2)$. Based on Thm.18, we deduct that the discrete dynamical system induced by $\varphi$ is chaotic in Li-Yorke sense when $\epsilon < \epsilon_0$ and $0 < \eta < 1/L$. $\qquad\square$

**Remark 20** (Beyond Li-Yorke Chaos). (*Thanks to valuable comments from Fryderyk Falniowski.*) Here the 3-periodic orbit of $\varphi$ can be used to establish a positive topological entropy [Misiurewicz, 2010], which implies not only Li-Yorke chaos but also distributional chaos, as well as the existence of a subsystem chaotic in the sense of Devaney [Li, 1993] (see e.g., Aulbach and Kieninger [2001], Falniowski et al. [2015] for their differences). So far these are only known in 1D though.

### B.3.4 On the Lyapunov exponent

*Proof of Thm.9.* All the norms for matrix in this proof is 2-norm (for simplicity, we omit its subscript).

Denoted by $\nu$ the invariant distribution of the deterministic map. Denote the special map where is $f_0 \equiv 0$ as $\varphi_0$:

$$\varphi_0(x) = x - \eta \nabla f_{1,\epsilon}(x)$$

With ergodicity, when $\epsilon \to 0$, we have

$$\lambda(x) = \lim_{n\to\infty} \frac{1}{n} \sum_{i=1}^{n} \ln \|\nabla\varphi|_{\varphi^{(i)}(x)}\|$$

$$= \int \ln \|\nabla\varphi|_x\| \, \nu(dx)$$

$$= \int \ln \|\nabla\varphi_0|_x + \eta \mathrm{Hess} f_0(x)\| \, \nu(dx)$$

Since $\mathrm{Hess} f_0$ is bounded, we know that

$$\lambda(x) = \int \ln \|\nabla\varphi_0|_x\| \, \nu(dx) + \mathcal{O}(\eta)$$

And then, we choose a bounded set $T$ and a mesh of which, denoted as $\Delta = \bigsqcup_{i\in\mathcal{I}} \Gamma_i$, $\forall \delta > 0$, we have $\mu$ is a simple function which is constant on each $Gamma_i$, where $\mathrm{supp}\mu \subset T$, $\int |\mu - \nu| \, dx < \delta$. Denoted the bound of $\epsilon\nabla^2 f_{1,\epsilon} = A$, then

$$\lambda(x) = \sum_{i\in\mathcal{I}} \int_{\Gamma_i} \ln \|\nabla\varphi_0|_x\| \, \nu(dx) + \mathcal{O}(\eta)$$

$$= \sum_{i\in\mathcal{I}} \int_{\Gamma_i} \ln \|\nabla\varphi_0|_x\| \, (\mu + (\nu - \mu)) dx + \mathcal{O}(\eta)$$

$$= \ln\left(\frac{\eta}{\epsilon}\right) + \sum_{i\in\mathcal{I}} \int_{\Gamma_i} \ln \|\epsilon\nabla^2 f_1(y)\| \, (\mu + (\nu - \mu)) dx + \mathcal{O}(\eta)$$

where $\sum_{i\in\mathcal{I}} \int_{\Gamma_i} \ln \|\nabla\varphi_0|_x\| \mu(dx) \to m$ and $\sum_{i\in\mathcal{I}} \int_{\Gamma_i} \ln \|\nabla\varphi_0|_x\| (\nu - \mu)(dx) < \delta A \to 0$. So we know that $\lambda(x) - \ln\left(\frac{\eta}{\epsilon}\right) \to m$ when $\epsilon \to 0$ first and then $\eta \to 0$. $\qquad\square$

**Remark 21.** Here we need $\varphi$ to be ergodic, which means the distribution of a single trajectory converges to the invariant distribution of the chaotic dynamical system. We don't have a reference, but please see section 3.1 for numerical test.

**Remark 22.** One may ask why $f_0$ doesn't appear in $m$. The reason is, the microstructure creates both local and global chaos, not the macrostructure; in fact, since $L \ll 1/\epsilon$, $L$ for the $L$-smooth $f_0$ gets absorbed in the high-order term in the proof.

**Remark 23.** When $f_1$ is periodic and $f_{1,\epsilon} = \epsilon f_1(x/\epsilon)$, we have an estimation of the order of convergence.

We divide the support of the invariant distribution into small parts according to the period of $\epsilon f_1(x/\epsilon)$, and enumerate them with $A_j, j \in \mathbb{N}$.

$$\lambda(x) = \sum_{i} \int_{A_j} \ln \|\nabla\varphi|x\| \, \nu(dx) + \mathcal{O}(\eta)$$

$$= \sum_{i} \int_{A_j} \frac{1}{\epsilon|\Gamma|} \left( \int_{\epsilon\Gamma} \ln \|\nabla^2 f_{1,\epsilon}(y)\| dy + \mathcal{O}(\epsilon) \right) \nu(dx) + \mathcal{O}(\eta)$$

$$= \ln\left(\frac{\eta}{\epsilon}\right) + \frac{1}{|\Gamma|} \int_{\Gamma} \ln \|\nabla^2 f_1(y)\| \, dy + \mathcal{O}(\epsilon + \eta)$$

$$= \ln\left(\frac{\eta}{\epsilon}\right) + m + \mathcal{O}(\epsilon + \eta).$$

## C    A possible origin of multiscale landscape from neural networks

It is possible that the (training) loss of a neural network satisfies the multiscale requirement of the presented theory. Here is an illustration in which multiscale training data together with periodic activation leads to a multiscale loss:

Consider the training of a 2-layer neural network to fit data $\{x^k, y^k\}_k$, where the output $y^k = y_0^k + y_1^k + \xi^k$ admits a decomposition into large scale behavior $y_0^k = g_0(x^k)$, microscopic detail $y_1^k = \epsilon g_1(\epsilon x^k)$, and i.i.d. noise $\xi_k$. Assume $g_0$ and $g_1$ are regular enough so that universal approximation (UA) works and they can be approximated by wide enough neural networks with $\mathcal{O}(1)$ weights. Consider MSE loss $\sum_k \|y^k - \sum_i a_i \sigma(W_i x^k + b_i)\|^2$ with $\sigma$ being the periodic activation in a recent progress [Sitzmann et al., 2020]. Then the loss admits a minimizer and in its neighborhood the loss satisfies Cond.1&2 for the following reason: omit $k$ without loss of generality, absorb bias into weight, and rewrite the loss as (denoting $\theta = [a_i, W_i]_i$)

$$f(\theta) = \left\| y_0 - \sum_{i \in I} a_i \sigma(W_i x) + \epsilon y_1 - \sum_{j \notin I} a_j \sigma(W_j x) \right\|^2 = \left\| g_0(x) - \sum_{i \in I} a_i \sigma(W_i x) \right\|^2$$

$$+ 2\epsilon \left\langle g_0(x) - \sum_{i \in I} a_i \sigma(W_i x), g_1(\epsilon x) - \sum_{j \notin I} a_j \sigma(W_j x) \right\rangle + \epsilon^2 \left\| g_1(\epsilon x) - \sum_{j \notin I} a_j \sigma(W_j x) \right\|^2$$

where $I$ and $I^c$ are sets of nodes, each large enough for UA to ensure vanishing loss. Renormalize by letting $\hat{x} = \epsilon x$ so that UA works for $g_1(\cdot)$, then the 2nd term rewrites as

$$2\epsilon \left\langle g_0(x) - \sum_{i \in I} a_i \sigma(W_i x), g_1(\hat{x}) - \sum_{j \notin I} a_j \sigma\left( \frac{W_j}{\epsilon} \hat{x} \right) \right\rangle.$$

This is in the form of $\epsilon \hat{f}_1(\theta/\epsilon, \theta)$ for some $\hat{f}_1(\phi, \varphi)$ that is quasiperiodic in $\phi$ (quasiperiodic because $\hat{x}$ is multi-dim). The 3rd term rewrites similarly. Thus, we see $f(\theta) = f_0(\theta) + f_{1,\epsilon}(\theta)$ where $f_0$ is the 1st term and $f_{1,\epsilon}(\theta) = \epsilon \hat{f}_1(\theta/\epsilon, \theta) + \epsilon^2 \hat{f}_2(\theta/\epsilon, \theta)$ for some $\hat{f}_1, \hat{f}_2$ quasiperiodic in the 1st argument. Such $f_{1,\epsilon}$ satisfies Cond.1&2 due to its quasiperiodic micro-scale. $\square$

# D More numerical evidence

## D.1 Period doubling

We illustrate numerically that $\varphi$, when viewed as a family of maps indexed by LR $\eta$, keeps under-going period doubling bifurcation as $\eta$ increases, and the period of $\eta$ eventually approaches infi-nite at a finite $\eta$ value, which is the chaos thresh-old (e.g., Alligood et al. [1997], Chap 11). This observation is rather robust to $f_0$, and we choose a convex but not strongly-convex example for an illustration.

The bifurcation diagram is plotted in Fig.9. For each $\eta$ value, we start with a fixed initial con-dition and iterate it using GD dynamics ($\varphi$) for sufficiently long so that the dynamics settle into an attractor, and then draw each of the thereafter iterations as a point on the diagram. For exam-ple, one can read from Fig.9 that there are two points at $\eta = 2.5\epsilon$, corresponding to an orbit of period 2. Although limited by the numerical resolution, one can see that the chaos threshold in this case is around $\eta \approx 3.5\epsilon$.

Figure 9: Bifurcation diagram of GD with $\epsilon = 10^{-3}$, $f_0 = x^4/4$ and $f_{1,\epsilon} = -\epsilon \cos(x/\epsilon)$.

Worth mentioning is that the chaos that first onsets is a local one, happening in a (and every) small potential well created by $f_{1,\epsilon}$. In other words, before global chaos for which LR is so large that GD can escape local well, arbitrarily large period already appears and chaos already onsets. This can be seen from Fig.9 as the boundaries of a small potential well, which is approximately $[-\epsilon\pi, \epsilon\pi]$, are marked by red dashed lines.

## D.2 A multi-dimensional demonstration

Our sufficient condition for chaos (Thm.8) is restricted to 1D problems, although our connection between $\varphi$ and $\hat{\varphi}$ limiting statistics (Sec.2.1) and the approximation of $\hat{\varphi}$ limiting statistics (Sec.2.2)

work for any finite dimension. We conjecture that stochasticity also appears in large LR GD for multidimensional multiscale objective functions. A numerical experiment consistent with this conjecture is presented, based on a classical strongly convex test function of Matyas:

Let $f_0$ be defined as

$$f_0(x, y) = 0.26(x^2 + y^2) + 0.48xy.$$

The small scale is arbitarily chosen to be

$$f_{1,\epsilon}(x, y) = \epsilon \sin(x/\epsilon) + \epsilon \cos(y/\epsilon), \epsilon = 10^{-7}.$$

The evolution of the empirical distribution of an ensemble, respectively under GD $\varphi$ and the stochastic map $\hat{\varphi}$, is shown in Fig.10, where good agreement is observed. The GD empirical distribution is also compared with rescaled Gibbs in Fig.11, where results again agree.

(a) Deterministic map

(b) Stochastic map

Figure 10: Comparison between the deterministic map and the stochastic map on Matyas function ($\eta = 0.01$) for testing Thm.4. Agreed histograms suggests that the limiting distributions of the two maps are close.

(a) $\eta = 0.1$

(b) $\eta = 0.01$

(c) $\eta = 0.001$

Figure 11: Test for the explicit expression of the invariant distribution. The surface is rescaled Gibbs and the histogram is the experiment result. They are overplotted after a rescaling by $\sqrt{\eta}$ in both axis. Obersved agreement is consistent with the rescaled Gibbs approximation.

In terms of deterministic chaos, although our sufficient condition for chaos (Thm.8) is only for 1-dim., the Lyapunov exponent estimate (Thm.9) works for any finite dimension as it assumes already ergodicity. Here we observe numerically that the deterministic map is chaotic and mixing (thus ergodic) despite of the $\geq 2$ dimension: see Fig.12 for the statistical behavior of a single orbit. A comparison with Fig.10 gives agreement in the statistics.

(a) Histogram of a trajectory      (b) x value of a trajectory      (c) y value of a trajectory

Figure 12: The histogram of a single trajectory. We can see that it is the same as the experimental result for the invariant distribution in Fig.10(b).

## D.3 Lyapunov exponent

Thm.9 provides a quantitative estimate of the Lyapunov exponent of the deterministic GD map $\varphi$. Although we required an additional strong convexity condition on $f_0$ for the geometric ergodicity of the stochastic map $\hat{\varphi}$, this result about the deterministic map does not have this requirement.

### D.3.1 On 1-dim periodic $f_{1,\epsilon}$

As an illustration, we pick multimodal nonconvex $f_0 = (x^2 - 1)^2$, together with $f_{1,\epsilon}(x) = \epsilon \sin\left(\frac{x}{\epsilon}\right)$. Fig.'s 13 and 14 respectively plot how the numerically computed Lyapunov exponent (computed by eq.4 with a random initial point) depends on $\eta$ (with fixed $\epsilon$) and on $\epsilon$ (with fixed $\eta$). The constant $m \approx \lambda(x) - \ln(\eta/\epsilon)$ is around 0.7 in both plots, which agrees with our theoretical estimate of $m = \frac{1}{2\pi} \int_0^{2\pi} \ln|\sin(y)|\,dy \approx -0.6931$.

(a) $\lambda(x)$ against $\eta$      (b) $\lambda(x) - \ln(\eta/\epsilon)$ against $\eta$

Figure 13: Dependence of the Lyapunov exponent on $\eta$

(a) $\lambda(x)$ against $\epsilon$          (b) $\lambda(x) - \ln(\eta/\epsilon)$ against $\epsilon$

Figure 14: Dependence of the Lyapunov exponent on $\epsilon$

### D.3.2    On 1-dim non-periodic $f_{1,\epsilon}$

The following experiment shows that Thm. 9 works for non-periodic $f_{1,\epsilon}$. Fig. 15 is the test on the quasiperiodic $f_{1,\epsilon}$ given in Fig. 5 and Example 2. The theoritical value for $m$ in Cond. 2 is $\lim_{n\to\infty} \int_0^n \ln|\sin(x) + 2\sin(\sqrt{2}x)|\, dx \approx -0.0117$, is the same as the experiment shows.

(a) $\lambda(x) - \ln(\eta/\epsilon)$ against $\eta$          (b) $\lambda(x) - \ln(\eta/\epsilon)$ against $\epsilon$

Figure 15: Dependence of the Lyapunov exponent on $\epsilon$ and $\eta$ for non-periodic $f_{1,\epsilon}$ (m=-0.0117).

### D.3.3    On the multi-dim case

Then we also test the theorem in a multi-dim case, whose $f_0$ is Matyas function and $f_{1,\epsilon}$ is periodic function, same as we did in Sec. D.2. We chose a random initial point, run sufficiently many iterations, and use eq.4 to compute it. At the same time, Thm.9 gives a theoretical estimation, with $m = \frac{1}{4\pi^2} \int_{[0,2\pi]^2} \ln\max(|\sin(x)|, |\cos(y)|)\, dx\, dy \approx -0.2669$. Fig.'s 16 and 17 show that this estimation, namely $\lambda(x) \approx m + \ln\left(\frac{\eta}{\epsilon}\right)$, is correct up to $\mathcal{O}(\epsilon + \eta)$ error.

(a) $\lambda(x)$ against $\eta$

(b) $\lambda(x) - \ln(\eta/\epsilon)$ against $\eta$

Figure 16: Dependence of $\lambda(x)$ on $\eta$ ($\epsilon = 0.00001$)

(a) $\lambda(x)$ against $\epsilon$

(b) $\lambda(x) - \ln(\eta/\epsilon)$ against $\epsilon$

Figure 17: Dependence of $\lambda(x)$ on $\epsilon$ ($\eta = 0.1$)

### D.4  Stochasticity of deterministic gradient descent with momentum

Just for illustrations, consider $f_0 = x^2/2$, $f_{1,\epsilon}(x) = \epsilon \sin(x/\epsilon)$, and two common ways for adding momentum:

#### D.4.1  Heavy ball

The iteration is [Polyak, 1964] $v_{n+1} = \gamma y_n - \eta \nabla f(x_n)$, $x_{n+1} = x_n + v_{n+1}$, with $v_0 = 0$. See the stochasticity of $x$ in Fig.18.

(a) Evolution of an ensemble

(b) Empirical distrib. of an orbit

(c) Iterations in an orbit

Figure 18: Heavy ball experiment. $\eta = 0.01$, $\epsilon = 0.0001$, and $\gamma = 0.9$.

### D.4.2 Nesterov Accelerated Gradient for strongly convex function (NAG-SC)

The iteration is [Nesterov, 2013] $y_{k+1} = x_k - \eta \nabla f(x_k)$, $x_{k+1} = y_{k+1} + c(y_{k+1} - y_k)$, with $y_0 = x_0$. $c = \frac{1-\sqrt{\mu\eta}}{1+\sqrt{\mu\eta}}$ where $\mu$ is supposed to be the strong convexity constant; we chose $\mu$ to be that for $f_0$, in this case $\mu = 1$. See the stochasticity of $x$ in Fig. 19. The smaller variance is due a different scaling for relating $\eta$ to a timestep in continuous time (see e.g., Su et al. [2014]).

(a) Evolution of an ensemble     (b) Empirical distrib. of an orbit     (c) Iterations in an orbit

Figure 19: NAG-SC experiment. $\eta = 0.01$, $\epsilon = 0.0001$.

### D.5 The nonconvex $f_0$ dichotomy: to escape or not to escape macroscopic potential well created by $f_0$?

What will happen when $f_0$ is nonconvex but multimodal? Both escapes from $f_0$'s local minima (and the corresponding potential wells) and nonescapes will be possible. Roughly speaking, it depends on how strong $f_{1,\epsilon}$ is when compared with $f_0$. Rmk.12 provided some discussions. To elaborate more, we first make a general remark:

**Remark 24.** As theoretically shown, especially in section 2.3.1, B.3.2 and 2.3.2, we see that chaos can be just a localized small-scale behavior, thus independent of the convexity of $f_0$. However, the limiting distribution of the deterministic map is a global property and it should depend on the global behavior of $f_0$. As explained in Rmk.12, when $f_0$ is not convex, it can happen that an orbit cannot jump between potential wells, and then unique ergodicity is lost in the sense that multiple ergodic foliations appear and respectively localize to individual potential wells. In this case, the limiting statistics is no longer unique. However, every connected subset of the support of an invariant distribution of the stochastic map can be an ergodic foliation, so if we regard the invariant distributions of the deterministic map and the stochastic map as convex combinations of the invariant distributions in each potential well, the conclusion in Theorem 4 still stands.

Then we demonstrate two possible outcomes concretely in numerical experiments. We will use the same test function, which is $f_0(x) = k(x^2 - 1)^2$ and $f_{1,\epsilon}(x) = \epsilon \sin(x/\epsilon)$. $x > 0$ and $x < 0$ are two potential wells of $f_0$.

We already obtained a bound on the relative strength between $f_0$ and $f_{1,\epsilon}$; it is $k_{critical} = \frac{3\sqrt{3}}{8}$ for whether the point can jump from one potential well to another. Fig.'s 20 and 21 respectively illustrates the long-time statistics of GD when $k = 0.05 < k_{critical}$ and $k = 5 < k_{critical}$. Results are consistent with theoretical predictions.

(a) Invariant distribution

(b) Histogram of a trajectory

(c) Histogram of another trajectory

(d) One trajectory

Figure 20: A non-convex mixing example. The initial condition is concentrated in the right potential well but barrier crossing happens. $k = 0.02$, $\eta = 0.05$ and $\epsilon = 0.0001$.

(a) One of the invariant distributions

(b) Histogram of a trajectory, starting in the right well

(c) Histogram of another trajectory, starting in the left well

(d) Landscape of $f_0$

Figure 21: A non-convex and non-mixing example. The initial condition is concentrated in the right potential well but no orbit can cross the potential barrier at $x = 0$. There is at least another invariant distribution in the left potential well due to symmetry. But if one restricts to the foliation within the potential well, convergence to a statistical limit still occurs. $k = 5$, $\eta = 0.05$ and $\epsilon = 0.0001$.

Interestingly, we observe that Rmk.15 still holds even though the orbit is confined in one potential well if $k$ is large. As $f''(1) > 0$, the function is strongly convex in a neighborhood of $x = 1$, and rescaled Gibbs can be approximated by a Gaussian density of $\exp(-16k(x - 1)^2)/Z$. Fig.22 shows that the ensemble empirical distribution indeed converges to this prediction as $\eta \to 0$.

(a) $\eta = 0.05$     (b) $\eta = 0.02$     (c) $\eta = 0.01$     (d) $\eta = 0.001$

Figure 22: Empirical distributions of a sufficiently evolved ensemble for different $\eta$ values when $k = 5$. The red line is the theoretical approximation in Rmk.15. Note x-axis has been zoomed in via $x \mapsto 1 + (x - 1)/\sqrt{\eta}$ for focusing on the essential part.