[Reviews · NeurIPS 2020]

Review 1

Summary and Contributions: This paper investigates the statistical behavior of applying GD to an objective function that can be written as two parts, a ``low frequency’’ part f_0 that gives the overall landscape and a ``high frequency’’ part f_{1,eps}, whose derivative behaves like a random variable in the small eps limit, independent of the input (Cond 1, 2). It is proved that when GD is applied to this objective, with a step size that maintains stability but does not resolve the variations of the high frequency component, the statistical behavior of GD can be approximated by that of a discrete-time stochastic process – GD with f_0 as the objective with a true noise term related to f_{1,\eps} . It is further demonstrated that the iterations exhibit chaotic behavior for certain ranges of learning rates, and in the limit of small learning rates the invariant distribution is closed to a Gibbs distribution of f_0, with variance given by f_1's statistics.

Strengths: There is some speculation from past work that GD with large learning rates gives a regularization effect. The strength of this paper is that some concrete statements on how that may occur are given, not directly on regularization but on an effectively noisy behavior that may arise from deterministic algorithms. I think this proposes another way of thinking about optimization in machine learning that couples the algorithm (GD) and the landscape (f_0 + f_1) that together may give rise to non-trivial behavior. I believe such a consideration is very important, as various interesting behavior that arises in machine learning today is possibly a mix of these factors. In this sense, I think the work is quite interesting and novel.

Weaknesses: The main weakness I can identify is the relevance of these results for machine learning. I think there appears to be some missing bridges that can be better explored. For example, the authors claim at several points that f_0 may represent the expected loss over a population of samples in a machine learning problem and f_1 the variations due to sample noise. However, it is not clear to me how finite samples will create f_1 that satisfies the given conditions. The paper will be improved if a direct connection can be made: what architectures/losses/distributions, when finitely sampled, will create the multi-scale behavior studied in this work, and in particular f_1 that satisfies Conditions 1,2? This will make the analysis more accessible and interesting for a machine learning audience. Another issue is with the actual content of the conditions, which I outline below in additional feedback.

Correctness: The theoretical results appear correct, although I did not check every result line by line.

Clarity: The exposition is generally clear, although some notation issues exist and the overall clarity is affected. See some detailed comments below.

Relation to Prior Work: Taking the appendix into account, the relation to prior work is sufficiently discussed.

Reproducibility: Yes

Additional Feedback: [After rebuttal] I appreciate the additional explanations in the rebuttal. I think the example (a more complete version) will go a long way in improving the paper, but as is presented I think not enough details is given for a proper evaluation, thus I look forward to reading a revised version of this work. Note that my tautology comment is not saying that the proof is trivial, but saying the way it is written masks the potential insights the proof may give, in particular, there should be a result that shows that such a limit in Cons 1 exists under some general conditions characterising the data and the model architecture. I believe the example provided in the rebuttal may potentially be useful for formalising this. --- Some detailed questions and comments: 1. Conditions 1 and 2. This follows from my comments on the main weakness. On first reading, these conditions appear not well-motivated. Concretely, from my understanding they essentially say that statistically, f_1’s derivatives behave like a random variable independent of the input in the limit of eps. Thus, the main results in this paper, which depends on these conditions, appears to say that: assuming they behave like random variables, then the GD process behaves like a stochastic process. Although proving this requires some technical work, fundamentally this sounds like a tautology, at least as currently written. Even if one cannot easily characterize how this effective behavior of f_1 may arise from data sampling, one can at least attempt to specify conditions on f_1 for which such a random variable \zeta exists in condition 1, instead of assuming such a limit exists as a condition. 2. Line 72: the second \nabla f_1 should be \nabla^2 f_1 3. Line 120: Theorem 3. I think it is useful to provide a definition of what chaotic, even a heuristic one. Otherwise, the content of this theorem is limited in the main text. 4. Line 190: Theorem 4. a. Notation P(R^d) is not defined b. Line 204: the limit points is with respect to topology induced by L^2 or some other metric? c. Can you give an example of \mathcal{F} and f that satisfy these conditions? This will improve readability 5. Theorem 7, Eq (3). Perhaps this can be written more clearly? It is not clear what \rho_\infty = \tilde {\rho} + O(\eta^2) in <some topology> actually means when these are probability density functions and do not form a vector space under usual addition.


Review 2

Summary and Contributions: UPDATE: I have read both author feedback and other referees' evaluations. (1) Isotropic $\zeta$: I agree with authors' responses. I just meant that it would be nice to have a form of distribution when $\zeta$ is distributed non-isotropically with general covariance as well. But this is very minor. (2) I really appreciated the neural-network experiments that show the claim of the paper, which strengthens its relevance to the neural-network community. I must confess that I am not entirely familiar with UCI Airfoil Self-Noise Data Set and am not sure why such a dataset is used (it would be even nicer to use more standard datasets like MNIST/CIFAR-10, if possible in the final edit phase), but I am fine with that. In light of (2), I increased my score from 6 to 7. -------- The current manuscript analyzes the stochasticity generated by deterministic gradient-descent (GD) algorithm, in an analogy with the chaotic/ergodic sampling in other deterministic systems.

Strengths: The idea of finding chaotic motions in deterministic GD is simple but novel. The claim is proved within some restricted setup and further illustrated with concrete toy loss landscapes. I believe that this is a theoretically interesting contribution.

Weaknesses: It is unclear whether or not the conditions used to prove the main claim (stochasticity in deterministic GD) are met in any practical setups. In addition, the assumption of isotropic noise (which leads to Gibbs) is far from being met in practice: for example in k-classification tasks, gradients often concentrate within the ~top-k eigenspace of Hessian. See, e.g., G. Gur-Ari, D. A. Roberts, E. Dyer, "Gradient Descent Happens in a Tiny Subspace." (I am not the author of this paper.)

Correctness: I believe that it is correct, although I have not checked every single detail or tried to replicate experiments.

Clarity: Overall the manuscript was clear but the presentation around lines 52 to 67 was hard to parse (e.g., it was not clear till later what epsilon is, what Gamma is, what the motivations behind specific choices of Conditions 1&2 are, etc.). That being said, the intuitive picture in Figure 1 with explanation around there was clarifying, which made it easy to digest the main claims of the manuscript. Experiments were further clarifying.

Relation to Prior Work: Relevant literature on large learning rates and chaos is properly cited in the section 1.2.

Reproducibility: Yes

Additional Feedback: Would it be possible to construct synthetic dataset for which the loss landscape of some model approximately satisfies the conditions used to prove the main claim (stochasticity in deterministic GD)? Better yet, is there any realistic dataset+neural-network architecture for which one can empirically show that the stochasticity can be observed in deterministic GD? (I understand that these suggestions are more for the future work but, if met, I am very willing to increase my score substantially.) Can one weaken the assumption of the isotropic noise (for which it was not surprising that one gets the Gibbs distribution)? This condition is typically not met in practice, as mentioned above. As alluded to above, it would be nice if the presentation around lines 52 to 67 is made clearer. Typo: in line 71 (and in line 72, the second inline equation), the \nabla on f_{1,\epsilon} should be \nabla^2.


Review 3

Summary and Contributions: In this work, the authors aim to understand the effect of setting large learning rates (LR) in (full)-Gradient Descent (GD) procedures. The main message of the paper is that, in a proper LR regime, the deterministic GD dynamics induces an asymptotic statistical distribution over the configurations which is similar to that reached by a described stochastic dynamics (rescaled Gibbs distribution). This is due to the fact that (when a multi-scale loss function is considered) one can observe the onset of local/global chaos when the LR exceeds certain values.

Strengths: Understanding the behavior of optimization algorithms in complex landscapes is a key step in building a theory of deep learning. The Dynamical Systems approach of the authors is interesting and allows the connection with powerful concepts and methods, that could prove useful also in future work.

Weaknesses: In my opinion, it is quite hard to understand the actual the scope of the presented results and the relevance of the conceptual implications in a typical machine learning setting (or even, in deep learning. This is where the authors seem to find the motivation for the present work, as described in the introduction). In particular, it is hard to separate hypotheses and intuitive claims from proven results, and it is not immediately clear what results are valid for multi-dimensional non-convex optimization problems. In my opinion, also the provided numerical evidence is not sufficient for clarifying this matter: I understand that, in a Dynamical Systems approach it is customary to consider one-dimensional problems, however how the 1-dim or 2-dim cases are related to high-dimensional dynamics remains obscure. Moreover, the authors liken some properties of large-LR GD with those of SGD, but then no actual comparison is offered (not even numerically). As mentioned, SGD is very important for achieving good generalization performance, and properly clarifying the relationship with large-LR GD would be very important. Finally, I don't completely agree with the choice of the authors of defining the behavior of large-LR GD "stochastic": I don't see the need of this, despite making for a good title.

Correctness: I think that the employed methods are correct, but many claims seem to remain(theoretically and numerically) unsupported.

Clarity: In my opinion, in the current version of the paper, the validity of the presented results and the working hypotheses considered by the authors are not presented straightforwardly. In general, I think the authors should rescale the scope of their discussion, focusing less on deep learning phenomena without providing proof of a concrete connection with them.

Relation to Prior Work: Yes, even though the deep learning context used to motivate the presented work feels a bit disconnected from the actual results presented in the paper.

Reproducibility: Yes

Additional Feedback: I think the authors should be more explicit in defining the range of validity of the assumptions behind their theory, and limit their unsupported claims. In my opinion, some numerical experiments in high-dimension (for example comparing SGD, large-LR GD and the stochastic dynamics) would drastically increase the impact of this work. Also, the mentioned parallel with SGD is interesting but completely unexplored. After rebuttal I believe that the authors made a real effort to try to include and address the many comments provided in the review process, even though I cannot verify this explicitly without the final version of the paper. I think that the modifications are in the right direction, strengthening the link with actual learning practices, and that describing additional experiments with more realistic settings will constitute a solid improvement of this work. I sincerely hope that the authors will also consider a comparison with actual SGD in the final version. I will therefore improve my score, from a 5 to a 6 (althought this work could potentally deserve a higher score).

[Author Response · NeurIPS 2020]

We appreciate the valuable comments, which urged us to embody explicit connections to practices of learning. Apology
that not all comments are replied here and our replies have to be short due to space, but they'll be fully addressed in a
revision. We plead a reconsideration based on the improvement, as our contribution is truly innovative and nontrivial.

**Re: connection to learning, and when Cond.1&2 hold.** Here is an example (simplified and only briefly explained
for length) in which the loss will be multiscale as considered in our paper: train a 2-layer neural network to fit data
$\{x^k, y^k\}_k$, where the output $y^k = y_0^k + y_1^k + \xi^k$ admits a decomposition into large scale behavior $y_0^k = g_0(x^k)$,
microscopic detail $y_1^k = \epsilon g_1(\epsilon x^k)$, and i.i.d. noise $\xi_k$. Assume $g_0$ and $g_1$ are regular enough so that universal
approximation (UA) works and they can be approximated by wide enough neural networks with $\mathcal{O}(1)$ weights. Consider
MSE loss $\sum_k \|y^k - \sum_i a_i \sigma(W_i x^k + b_i)\|^2$ with $\sigma$ being the periodic activation in a recent progress [Implicit Neural
Representations with Periodic Activation Functions, 2020]. Then there exists a minimizer and in its neighborhood the
loss satisfies Cond.1& 2: omit $k$ WLOG, absorb bias into weight, and rewrite the loss as (denote by $\theta = [a_i, W_i]_i$)

$$f(\theta) = \left\| y_0 - \sum_{i \in I} a_i \sigma(W_i x) + \epsilon y_1 - \sum_{j \notin I} a_j \sigma(W_j x) \right\|^2 = \left\| g_0(x) - \sum_{i \in I} a_i \sigma(W_i x) \right\|^2$$

$$+ 2\epsilon \left\langle g_0(x) - \sum_{i \in I} a_i \sigma(W_i x), g_1(\epsilon x) - \sum_{j \notin I} a_j \sigma(W_j x) \right\rangle + \epsilon^2 \left\| g_1(\epsilon x) - \sum_{j \notin I} a_j \sigma(W_j x) \right\|^2$$

where $I$ and $I^c$ are sets of nodes, each large enough for UA to ensure vanishing loss. Renormalize by letting $\hat{x} = \epsilon x$ so
that UA works for $g_1(\cdot)$, then the 2nd term rewrites as

$$2\epsilon \left\langle g_0(x) - \sum_{i \in I} a_i \sigma(W_i x), g_1(\hat{x}) - \sum_{j \notin I} a_j \sigma\left( \frac{W_j}{\epsilon} \hat{x} \right) \right\rangle.$$

This is in the form of $\epsilon \hat{f}_1(\theta/\epsilon, \theta)$ for some $\hat{f}_1(\phi, \varphi)$ that is quasiperiodic in $\phi$ (quasiperiodic because $\hat{x}$ is multi-
dim). The 3rd term rewrites similarly. Thus, we see $f(\theta) = f_0(\theta) + f_{1,\epsilon}(\theta)$ where $f_0$ is the 1st term and $f_{1,\epsilon}(\theta) =$
$\epsilon \hat{f}_1(\theta/\epsilon, \theta) + \epsilon^2 \hat{f}_2(\theta/\epsilon, \theta)$ for some $\hat{f}_1, \hat{f}_2$ quasiperiodic in the 1st argument. Such $f_{1,\epsilon}$ satisfies Cond.1&2 due to its
quasiperiodic small scale. □

Like most theory papers, we also present numerical experiments in which our conclusions still hold although conditions
for our theorems no longer apply. Thanks to the reviews the following will be added (and expanded):

**Neural network training.** We use fully connected 5-16-2 MLP to regress UCI Airfoil Self-Noise Data Set, with leaky
ReLU, MSE as loss, and batch gradient. Fig.1 shows large LR again produces stochasticity as our paper studies.

(a) loss histogram  (b) loss orbit  (c) hist of a weight parameter  (d) orbit of a weight parameter

(e) hist of a bias parameter  (f) orbit of a bias parameter  (g) loss orbit (small LR)  (h) parameter orbit (small LR)

Figure 1: (a)-(f) use LR=0.0165 (large) and demonstrate stochasticity originated from chaos as GD converges to a statistical distribution rather than a local min. (g,h) use LR=0.001 (small) and GD converges to a local min.

**Re: $f_{1,\epsilon}$ satisfying Cond.1&2 is like a random variable; tautology?** $f_{1,\epsilon}$ does contribute like a r.v., but this needs to
be proved, which is one of our main contributions – note both $x$ and $f_{1,\epsilon}(x)$ are deterministic even under Cond.1&2!
Cond.1&2 use auxiliary random variables to define the needed $f_{1,\epsilon}$, but $f_{1,\epsilon}$ is a deterministic function.

**Re: weaken isotropic noise assumption?** We don't require isotropic 'noise'. Kindly see e.g., Thm.2, which contains
2 statements: (i) convergence to stochastic behavior for general covariance; (ii) explicit characterization of the limiting
statistics when covariance is isotropic (note the same thing holds for SGD).

**Re: valid in multi-dim?** Apology that multi-dim. and nonconvex demonstrations were left in Appendix C.2, C.3.3, &
C.5. This rebuttal also adds a neural network example, which is high-dim. & nonconvex, and our conclusion still holds.

[Meta-Review · NeurIPS 2020]

Before the author response, all the reviewers seem agree that the results were quite interesting (and I agree), but had a concern about the connection to ML. The author response included examples which mostly addressed this concern, so two reviewers recommended acceptance, while another (reviewer 1) recommended rejection, but was borderline. However, I feel the remaining concerns by reviewer 1 are rather minor.